**Measurement report: Crustal materials play an increasing role in elevating particle pH: Insights from 12-year records in a typical inland city of China.**

Hongyu Zhang[1,2], Shenbo Wang[2,3]*, Zhangsen Dong[1,2]*, Xiao Li[2,3], Ruiqin Zhang[2,3]

[1] Collage of Chemistry, Zhengzhou University, Zhengzhou, 450000, China

[2] Research Institute of Environmental Sciences, Zhengzhou University, Zhengzhou 450000, China

[3] School of Ecology and Environment, Zhengzhou University, Zhengzhou, 450000, China

* Corresponding authors: Shenbo Wang and Zhangsen Dong

E-mail address: shbwang@zzu.edu.cn and dzszzu1990@163.com

## Abstract

Particle acidity serves as a key determinant in atmospheric chemical processes. Emerging concerns regarding aerosol acidity trends have been highlighted amid China's sustained initiatives to mitigate emissions of both acidic and alkaline precursors, especially in North China, which is significantly affected by dust aerosol. 12-year observational data in Zhengzhou reveal that the annual average $PM_{2.5}$ concentration decreased from $162 \pm 81$ $\mu g/m^3$ in 2011 to $60 \pm 41$ $\mu g/m^3$ in 2022, with the largest reduction in sulfate (73%). Correspondingly, the annual particle pH increased by 0.10 units from 2011 to 2019. In addition, the elevated particle pH in 2015 and 2018 was notably influenced by the increase in $TNH_x$ ($NH_3 + NH_4^+$). Note that the crustal material concentrations and their proportions increased significantly during 2019–2022, which might be responsible for the resuspension of surrounding soil dust. Even though the $TNH_x$ concentration was decreasing, the annual average growth rate of pH values increased to 0.21 units from 2019 to 2022. This phenomenon is not unique to Zhengzhou, as major cities in the North China Plain have also experienced a pronounced upward trend in coarse particles after 2019. Therefore, the long-term evolution of particle acidity in North China will require comprehensive consideration of synergistic effects involving acidic precursors, ammonia, and crustal materials.

**Keywords:** Dust, aerosol acidity, sources, North China Plain, control measurement

**Synopsis:** The future ammonia reduction policies in North China may not lead to a rapid increase in particle acidity in the presence of crustal materials., which further elevated the particle pH after 2019.

## Graphical abstract:

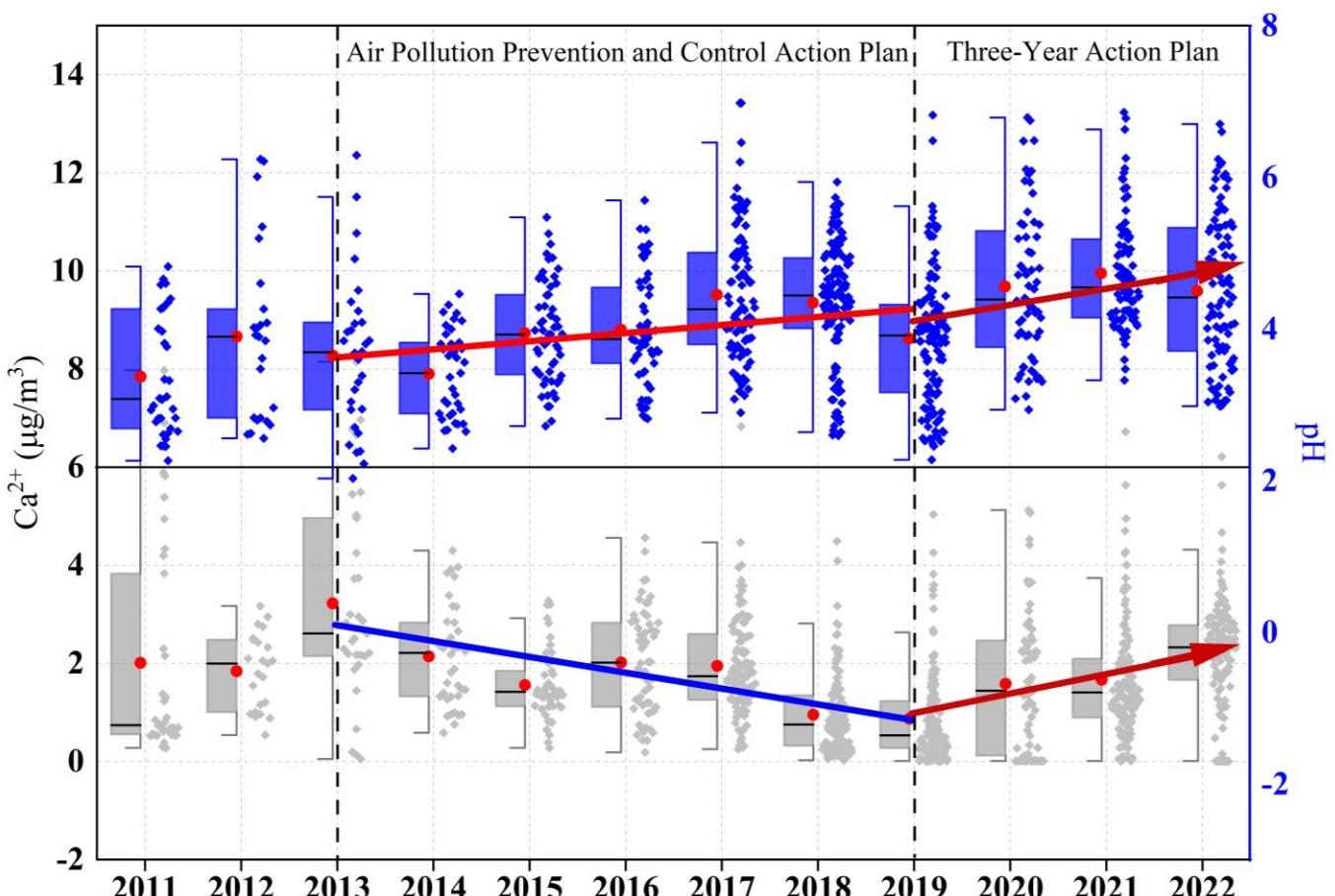

## Highlights:

- Crustal material concentrations and their proportions increased significantly during 2019–2022;
- The resuspension of surrounding soil dust may determine the rebound of crustal material concentrations;
- Rebound in crustal material further elevated the particle pH.

# 1 Introduction

Particle acidity is a critical parameter that affects atmospheric chemistry, such as the gas-particle partitioning of semi-volatile and volatile species (Surratt et al., 2010; Guo et al., 2016), the solubility of metals (Tao and Murphy, 2019), acid-catalyzed reactions (Rengarajan et al., 2011), and acid deposition (Mao et al., 2009), thereby determining aerosol concentration and chemical composition, as well as impacting human health, ecosystems, and climate (Li et al., 2017; Pye et al., 2020; Su et al., 2020; Nenes et al., 2021). Generally, the global fine particulate matter (PM$_{2.5}$, aerodynamic diameter ≤ 2.5 μm) exhibits a bimodal pH distribution ranging from 1–3 (e.g., in the United States and Europe) (Guo et al., 2015; Battaglia et al., 2017; Masiol et al., 2020; Zhang et al., 2021) and 4–5 (e.g., in East Asia) (Kim et al., 2022; Sharma et al., 2022). The atmosphere rich in gaseous ammonia (NH$_3$) and crustal material (CM) shows significant pH buffering effects (Wang et al., 2020; Zheng et al., 2020; Karydis et al., 2021), which is a dominant factor that drives the high particle pH in East Asia (Karydis et al., 2021; Zhang et al., 2021; Kim et al., 2022; Sharma et al., 2022).

In recent years, the changing trends in particle pH have become a research focus, especially in China, in response to air pollution control policies, i.e. Air Pollution Prevention and Control Action Plan (2013–2018) and Three-Year Action Plan (2018–2020). The annual average PM$_{2.5}$ concentration in Beijing dropped by 64% from 89.5 μg/m³ in 2013 to 32 μg/m³ in 2023 (MEP, 2023), with a clear downward trend of sulfate concentration, and nitrate surpassing sulfate as the primary component (Zhai et al., 2019; Zhou et al., 2019; Li et al., 2023). The atmospheric behavior of ammonium, governed by gas-particle partitioning processes involving ammonia (NH$_3$) as the predominant alkaline gas,

demonstrates notable stability in concentration levels, with observational records showing less than 5%

interannual variation in $NH_3$ column densities over North China during 2015–2019 (Dong et al., 2023).

Under such conditions, the dominant inorganic aerosol component transitions from ammonium sulfate

to ammonium nitrate. This compositional shift enhances atmospheric particulate hygroscopicity due

to ammonium nitrate's superior water uptake capability, ultimately elevating particle pH levels through

aqueous-phase dilution mechanisms (Wexler and Seinfeld, 1991; Pinder et al., 2007, 2008; Heald et

al., 2012; Weber et al., 2016). For instance, a significant increase in the nitrate-to-sulfate molar ratio

from 2014–2017 in Beijing resulted in the particle pH increasing from 4.4 to 5.4 (Xie et al., 2020).

Moreover, increased $NH_3$ concentrations raised particle pH by 0.3–0.4 units from 2014/2015 to

2018/2019 in Beijing (Song et al., 2019). Over Europe and North America, the pH has increased

strongly from about 2.8 and 2.2 during the 1970s to 3.9 and 3.3 in 2020 respectively, especially during

the 1990s, with significantly increasing $NH_3$ emission (Karydis et al., 2021). On the contrary, modeling

results indicate a continuous decline in pH in East Asia from 1970 to 2020 due to sharp increases in

$SO_2$ and $NO_x$ emissions (Karydis et al., 2021). In addition, the $PM_{2.5}$ pH showed a slight decrease of

0.13 from 2018 to 2022 summer in Beijing due to the change in total nitrate ($NO_3^- + HNO_3$) (Li et al.,

2023). Moreover, Zhou et al. (2022) found a decreasing pH trend from 2011 to 2019 in eastern China,

primarily influenced by temperature, followed by sulfate and non-volatile cations. Similarly, Nah et al.

(2023) observed a decreasing pH trend from 2011 to 2020 in Hong Kong, attributing it to temperature

and sulfate levels. Thus, concerns have been raised about the potential increase in the acidity of aerosol

and precipitation due to China's ongoing efforts to reduce ammonia emissions, which pose severe

health risks and acid deposition (Liu et al., 2019; Shi al., 2019).

85 In addition to $NH_3$, CM is another key alkaline substance, that buffers particle pH. $Ca^{2+}$ can form

86 insoluble $CaSO_4$ with sulfate, reducing sulfate concentration in the aqueous phase of aerosol, and thus

87 lowering $H^+$ and aerosol liquid water content (ALWC) concentrations and enhancing particle pH (Ding

88 et al., 2019; Karydis et al., 2021). Moreover, non-volatile cations can lower the molar ratio of ammonia

89 to sulfate, leading to an increase in particle pH (Zheng et al., 2022). Karydis et al. (2021) framework

90 demonstrated that CM played a critical buffering role in sustaining aerosol pH around 7 across the

91 Middle East arid regions. The model sensitivity tests revealed that under hypothetical dust-free

92 conditions (CM = 0), aerosol acidity would escalate to pH~4 due to $NH_4^+/SO_4^{2-}$ domination. Wang et

93 al. (2022) reported that non-volatile cations accounted for approximately 8–17% of hourly aerosol pH

94 variation. Li et al. (2023) indicated that the buffering effect of cations was the major reason for the

95 relatively small pH changes from 2018 to 2022 in Beijing, emphasizing that reducing coarse particle

96 emissions in the future could significantly decrease particle pH. In addition, there was a rising trend in

97 the contribution of CM to particle pH in Tianjin, China (Shi et al., 2017). Therefore, it is evident that

98 CM has a significant impact on the variation of particle pH, especially in North China, which is

99 significantly affected by dust aerosol, but the trend of CM concentration and its long-term implication

100 is still lacking unfortunately.

101 Zhengzhou presents unique atmospheric chemistry that distinguishes it from other mega-cities in

102 North China. As the capital of China's foremost agricultural province (Henan Province, contributing

103 18% of national $NH_3$ emissions), Zhengzhou's $PM_{2.5}$ composition combined substantial crustal

104 material ($15 \pm 3\%$ in $PM_{2.5}$ vs. <10% in Beijing) with exceptional ammonia abundance (Huang et al.,

105 2012; Liu et al., 2018; Wang et al., 2018). This created distinct particle acidity characteristics,

maintaining pH 4.5–6.0 compared to lower pH levels (3.3–5.4) in other cities like Beijing (Ding et
al.,2019; Zhang et al., 2021). However, two critical research gaps persist: (1) the long-term evolution
of CM under control policies remains unquantified; (2) the role of CM on pH buffer capacity in $NH_3$-
enriched environments lacks systematic assessment.
To address these gaps, our study pioneers the first multi-decadal analysis (2011–2022) coupling
$PM_{2.5}$ components with thermodynamic modeling through three key innovations: (1) revealing the
long-term trends of CM, (2) analyzing the variations of CM sources, and (3) exploring pH trend and
its relationship with CM. The resultant findings advance our understanding of urban aerosol acidity
chemistry by underscoring the critical role of CM.

## 115  2 Experiment and method

### 116  2.1 Instruments and Measurements

Sampling was conducted on the fourth-floor platform at Zhengzhou University (34.75° N, 113.61°
E) in Zhengzhou, China. The sampling site (Fig. S1), approximately 14 m above the ground, is
primarily surrounded by residential areas with well-developed transportation networks and no
significant industrial sources. There are two highways located 3 km to the south and 7 km to the east.
Additionally, a coal-fired power plant located 6 km to the east was shut down in 2020, and a gas-fired
power plant is situated 3 km to the south.
Samples were collected using a high-volume sampler (TE-6070D, Tisch, USA) and air particulate
samplers (TH-16A, Tianhong, China) from April 2011 to December 2022. Two quartz filters and two
Teflon filters were used daily from 10:00 AM to 9:00 AM the next day, resulting in a total of 5848
samples. After excluding abnormal data due to instrument malfunctions, 4228 valid samples were
obtained. Detailed information on the samples is provided in Table S1. Organic carbon (OC) and
elemental carbon (EC) were analyzed using a carbon analyzer (Model 5L, Sunset Laboratory, USA).
Water-soluble inorganic ions ($Cl^-$, $NO_3^-$, $SO_4^{2-}$, $Na^+$, $NH_4^+$, $K^+$, $Mg^{2+}$, and $Ca^{2+}$) were measured using
ion chromatography (ICS-90 and ICS-900 models, Dionex, USA) (Yu et al., 2017; Jiang et al., 2018).
Elements were analyzed using a wavelength dispersive X-ray fluorescence spectrometer (S8 TIGER,
Bruker, Germany) to determine concentrations of Fe, Na, Mg, Al, Si, Cl, K, Ca, V, Ni, Cu, Zn, Cr, Mn,
Co, Cd, Ga, As, Se, Sr, Sn, Sb, Ba, and Pb (Tremper et al., 2018). Meteorological conditions, including
temperature (T), relative humidity (RH), and wind speed (WS) were obtained using an automatic
weather station (Wang et al., 2019). Blank filters were also routinely analyzed with each batch of
samples to detect sample contamination and provide quality assurance on the elemental concentrations.
Detailed analytical methods and quality control are described in the supplement (Text S1). The method
detection limits and measurement uncertainties are summarized in Table S2. The quality assurance
protocol excluded temporally discrete dust storm and precipitation periods to prevent contamination
of the source analysis of CM and modeling particle pH, given that such events induce non-
representative extremes in both crustal element concentrations and pH values, coupled with elevated
PM measurement uncertainties. The annual mean $PM_{2.5}$ concentration data for cities in the North China
Plain were obtained from the China National Environmental Monitoring Center (CNEMC), available
at https://www.cnemc.cn/.

## 2.2 Data Analysis

### 2.2.1 Mass reconstruction

The calculation method for CM is as follows (Tian et al., 2016):

$$[CM] = 1.89 \times [Al] + 2.14 \times [Si] + 1.4 \times [Ca] + 1.43 \times [Fe] + 1.94[Ti] \tag{1}$$

where [Al], [Si], [Ca], [Fe] and [Ti] represent the concentrations of the respective elements ($\mu g/m^3$), but Ti was not measured.

### 2.2.2 Thermodynamic model

The particle pH was calculated using the ISORROPIA-II mode (version 2.1, http://isorropia.eas.gatech.edu). The input data (excluding RH ≤ 30%), including $SO_4^{2-}$, $TNO_3$ ($HNO_3$ + $NO_3^-$), $TNH_x$ ($NH_3$+$NH_4^+$), $Ca^{2+}$, $K^+$, $Na^+$, $Mg^{2+}$, $Cl^-$, RH and T, with the temporal resolution aligned with the sampling periods (from 10:00 AM to 9:00 AM the following day). Input data (excluding RH ≤ 30%) included $SO_4^{2-}$, $TNO_3$ ($HNO_3$ + $NO_3^-$), $TNH_x$ ($NH_3$+$NH_4^+$), $Ca^{2+}$, $K^+$, $Na^+$, $Mg^{2+}$, $Cl^-$, RH and T. The concentrations of hydrogen ions in air ($H_{air}^+$) and ALWC were calculated using the aerosol equilibrium composition system $Na^+$-$K^+$-$Ca^{2+}$-$Mg^{2+}$-$NH_4^+$-$SO_4^{2-}$-$NO_3^-$-$Cl^-$-$H_2O$ $H_{air}^+$ (Fountoukis and Nenes, 2007). pH values were calculated using the following formula:

$$pH = -\log_{10} H_{aq}^+ \cong -\log_{10} \frac{1000 H_{air}^+}{ALWC_i + ALWC_o} \cong -\log_{10} \frac{1000 H_{air}^+}{ALWC_i} \tag{2}$$

$$ALWC_o = \frac{m_{org} \rho_w}{\rho_w} \frac{\kappa_{org}}{\left(\frac{1}{RH} - 1\right)} \tag{3}$$

where $ALWC_i$ and $ALWC_o$ refer to the ALWC for inorganic and organic components, respectively.

$m_{org}$ denotes the mass of organic aerosol, $\rho_w$ is the density of water (1.0 g/cm$^3$), $\rho_{org}$ is the density of
organic material (1.4 g/cm$^3$) (Guo et al., 2015), $k_{org}$ is the hygroscopicity parameter for organic aerosol
(0.087) (Chang et al., 2010; Li et al., 2016). The ISORROPIA-II model operated under metastable
conditions in the forward mode. Due to the lack of measured data for gaseous HNO$_3$ and NH$_3$, TNO$_3$
was represented solely by NO$_3^-$. The concentration of NH$_3$ was simulated based on a linear regression
equation proposed by Wei et al. (2023), who used the same data as this study from 2013 to 2020:
$$NH_3 = 19.909 \times RH + 0.559 \times T - 0.35 \times NH_4^+ + 0.123 \times NO_3^- + 2.159 \times Cl^- - 0.224 \times SO_4^{2-} - 154.923 \quad (4)$$
where NO$_3^-$, SO$_4^{2-}$, NH$_4^+$, and Cl$^-$ correspond to their respective concentrations (μg/m$^3$). To validate
the applicability of Equation 4 for annual NH$_3$ estimation and pH simulation in Zhengzhou, this study
utilized both observed NH$_3$ data (from a Thermo Scientific URG-9000D ambient ion monitor, USA)
and calculated NH$_3$ values derived from Equation 4 at the same monitoring site throughout 2022,
inputting them into the thermodynamic model for pH simulation. As shown in Figure S2, pH values
calculated from observed and simulated NH$_3$ exhibit good agreement ($r = 0.97$, $P < 0.01$). Furthermore,
NH$_3$ concentrations modeled by ISORROPIA demonstrate a significant correlation ($r = 0.95$, $P < 0.01$)
with that simulated NH$_3$ by Equation 4. These results collectively demonstrate the reliability of the
NH$_3$ estimation method in this study.
**2.2.3 HYSPLIT analysis**
Backward trajectories were calculated using the mixed-particle Lagrangian integrated trajectory
method (HYSPLIT, https:// www.ready.noaa.gov/HYSPLIT_traj.php). Meteorological input data were
from the Global Data Assimilation System (GDAS) with 3D wind vectors, temperature, relative
humidity, geopotential height, surface pressure, and boundary layer diagnostics. 24-h backward
trajectories were simulated for air masses above 1000 m above ground level in Zhengzhou. While the
surface elevation of Zhengzhou is approximately 100 m above sea level (ASL), setting the height at
1000 m ASL takes into account the minimum altitude needed to traverse the average elevation of the
Taihang Mountains (ranging from 1000 to 1500 m ASL). This ensures that the simulated trajectory
paths over this topographical barrier are physically realistic.
The Angle Distance algorithm was used to cluster air mass trajectories, enabling the identification
of dominant air mass directions and potential pollution sources affecting the study site during different
periods. The optimal number of clusters was determined by evaluating the spatial variance (SPVAR)
of each trajectory from the cluster mean and the total spatial variance (TSV). The final classification
was selected just before the second rapid increase in TSV. The underlying principle is that TSV initially
rises sharply during clustering, then increases gradually; however, once the number of clusters reaches
a certain threshold, TSV surges again, indicating that the merged clusters are highly dissimilar, marking
the end of the classification process. The classification results correspond to the different air mass
categories before this final merging step. The mean trajectories of these clusters represent the primary
airflow patterns at the target site during the analysis period (Wang et al., 2009). Subsequently,
trajectories from two periods, 2013–2018 and 2019–2022, were clustered separately to analyze the
variations between the two policy implementation periods.

# 3 Results and discussion

## 3.1 Temporal variations in chemical components

Over the past twelve years, the Chinese government implemented the Air Pollution Prevention and Control Action Plans (2013–2018) and the Three-Year Action Plan (2018–2020). The Air Pollution Prevention and Control Action Plan focused on reducing $PM_{2.5}$ concentrations in key regions and aiming to cut $PM_{2.5}$ levels by 10–25% in priority areas over five years. To achieve these goals, it adopted several measures. In terms of industrial restructuring, it mandated the elimination of a large amount of outdated production capacity in industries such as iron/steel and cement to optimize the industrial structure and reduce high-pollution production. For emission standards, it set strict requirements for multiple industrial sectors, especially coal-fired power plants, and gradually introduced ultra-low emission requirements to control pollutants like $SO_2$, $NO_x$, and PM. Regarding energy transition, it promoted a shift from coal to cleaner energy sources, including capping coal consumption in certain regions and restricting the construction of small-scale coal-fired boilers. Subsequently, the Three-Year Action Plan was carried out to continue improving air quality with a broader scope of regions under control, further reducing pollutant emissions and enhancing the overall air quality index. The measures included enhanced transportation controls, such as introducing stricter vehicle emission standards (like National VI standards for vehicles) and establishing diesel truck exclusion zones in many cities to reduce emissions from the transportation sector. It also adopted precision governance through grid-based environmental supervision, dividing areas into small grids for more accurate and efficient monitoring of pollution sources. Additionally, it strengthened the legal

and institutional framework by revising relevant laws, such as the Air Pollution Prevention and Control
Law, to strengthen legal penalties for environmental violations and implementing an environmental
tax system to encourage enterprises to reduce emissions.

224        The long-term trends in $PM_{2.5}$ concentrations and its chemical components in Zhengzhou from

2011 to 2022 are depicted in Fig. 1, with annual average concentrations listed in Table 1.
Correspondingly, the annual average concentration of $PM_{2.5}$ in Zhengzhou decreased from $162 \pm 81$
$\mu g/m^3$ in 2011 to $60 \pm 41$ $\mu g/m^3$ in 2022, representing a reduction of approximately 63%. In particular,
the reduction rate reached 72% after 2013. As for chemical components, the largest reductions were
observed in $SO_4^{2-}$ (79%), decreasing from $38.0 \pm 19.9$ $\mu g/m^3$ in 2013 to $7.9 \pm 4.5$ $\mu g/m^3$ in 2022,
followed by EC (76%). Additionally, the concentrations of $NH_4^+$ and $NO_3^-$ also significantly decreased
by 68% and 56%, respectively. The proportion of each component in $PM_{2.5}$ (Fig. S3) reveals a decrease
in $SO_4^{2-}$, $K^+$, and $Cl^-$, indicating effective control measures targeting coal and biomass combustion
(Lei et al., 2021). However, the proportions of $NO_3^-$ and OC in $PM_{2.5}$ rose from 11% and 12% in 2013
to 13% and 17% in 2022, respectively, similar to the trend observed in the North China Plain (Wen et
al., 2018; Zhai et al., 2019; Li et al., 2023).
**3.2 Temporal variations in CM**

237        Notably, there is no clear declining trend in the CM concentration, with a rebound observed during

2020–2022 (Fig. 1i). Furthermore, the proportion of CM in $PM_{2.5}$ exhibits a significant upward trend
(Fig. S3). To further analyze its trend, sampling data were divided into three periods corresponding to
governmental stages: 2011–2013, when no special control measures were implemented; 2013–2019,
coinciding with the implementation of the Air Pollution Prevention and Control Action Plan; and
2019–2022, coinciding with the Three-Year Action Plan. During these periods, Henan Province and
Zhengzhou City implemented several dust control policies summarized in Table S3. As shown in Fig.
2a and 2b, the mass concentration of CM peaked at $14.6 \pm 8.3$ μg/m$^3$ in 2013, accounting for 8% of
PM$_{2.5}$. To evaluate the inter - annual change trend of CM, the Mann - Kendall method, Sen's slope,
and Least - Squares (LS) slope were comprehensively used with the results presented in Table S4.
From 2013 to 2019, the CM concentration notably decreased from $14.6 \pm 8.3$ to $8.5 \pm 7.8$ μg/m$^3$, with
an annual average decline rate of 0.81 μg/(m$^3$·year) from LS slope [0.015 μg/(m$^3$·year) from Sen's
slope]. Apart from control measures, the interannual meteorological analysis shows (Fig. S4) WS
exhibited a declining trend, with a decrease rate of 43%, while RH showed an increasing trend at a rate
of 8% from 2013 to 2019, under which conditions that were unfavorable for dust resuspension (Wang
et al., 2013, 2018). Seasonal trends (Fig. S5) reveal significant declines in CM during spring in 2013–
2019 with WS decreasing from 2.2 m/s in 2013 to 1.4 m/s in 2019 (Fig. S6) and stable RH (Fig. S7).
Similarly, summer CM reductions in 2013–2019 corresponded with WS declines. These patterns
suggest spring-summer CM improvements resulted from the synergistic effects of meteorological
changes and dust control policies. Conversely, autumn-winter seasons showed limited CM reductions
despite comparable WS decreases in 2013–2019, highlighting the need for enhanced dust emission
controls in Zhengzhou during these seasons. As for the individual crustal elements in Fig. S8, Ca
exhibited the highest average annual decline rate of 33% during 2013–2019, followed by Al. Si showed
a less pronounced decline, attributed to its association with soil dust, where control measures for
exposed soil are lacking (Zhang et al., 2020). In addition, the Ca$^{2+}$ concentration as depicted in Fig. 2c
decreased from $3.2 \pm 2.1$ μg/m³ in 2013 to $2.2 \pm 1.1$ μg/m³ in 2019, with an approximate annual average
decline rate of 0.32 μg/(m³·year) from LS slope [4.14E–03 μg/(m³·year) from Sen's slope] in Table
S4, further demonstrating the decline in dust source. It was worth noting that the proportions of CM,
Ca, Al, Fe, Si, and $Ca^{2+}$ in $PM_{2.5}$ have shown consecutive annual increases from 2013 to 2019, with
CM proportion increasing from 8% in 2013 to 14% in 2019, indicating that CM reduction lagged
behind $PM_{2.5}$ reduction efforts in Zhengzhou during this period. Additionally, both concentration and
proportion of $Ca^{2+}$ in 2022 ($2.2 \pm 1.1$ μg/m³ and 14%) were higher than in other cities of China, such
as Beijing (1.0 μg/m³ and 2.8%), Tianjin (0.5 μg/m³ and 1.4%), and Xiamen (0.48 μg/m³ and 1.5%)
(Shi et al., 2017; Xu et al., 2025; Zhang et al., 2021). These results indicate that CM remained an
important component of $PM_{2.5}$ in Zhengzhou City.
During 2019–2022, both CM and $Ca^{2+}$ concentrations exhibited significant rebounds, with annual
growth rates of 0.24 and 0.4 μg/(m³·year) from LS slope [5.80E–03 and 5.42E–03 μg/(m³·year) from
Sen's slope], respectively, and their proportions increased from 14% and 2% in 2019 to 22% and 5%
in 2022. CM concentrations rebounded in all seasons, particularly in winter (Fig. S5). Changes in
meteorological conditions may be a significant factor contributing to these concentration rebounds,
accompanied by the average WS increased by 0.14 m/s and RH decreased by 7% from 2020 to 2022
(Fig. S4, S6, and S7), facilitating dust resuspension. Furthermore, the lack of more effective dust
control measures, as indicated by the absence of significant changes in the dust control policies from
the Air Pollution Prevention and Control Action Plan and Three-Year Action Plan, may be another
important factor contributing to the rebound of dust.

## 3.3 Sources of CM

Elemental ratios were employed to characterize the sources of CM, with the Ca/Al ratio widely recognized as a reliable indicator of sandy origin (Zhang et al., 2017). In addition, significant variations in Ca/Si ratios (Table S5) were observed among different dust sources (Road, Construction, Piles, Soil). Fig. 3a illustrates the trend in Ca/Si ratios from 2011 to 2022. After 2013, Ca/Si ratios showed a declining trend annually, with the average ratio decreasing from a peak of 1.6 in 2016 to a lowest of 0.4 in 2022. Compared with Ca/Si ratios from different types of dust sources, the effect of road and construction dust on CM has gradually decreased. This may be attributed to the implementation of dust control measures such as enclosure, shielding, and dust suppression at construction and demolition sites, as well as dust control on ground surfaces and roads (Table S5). During 2019–2022, the average Ca/Si ratio remained below 1, with a mean of 0.4 in 2022, indicating that soil dust predominantly contributed to CM. Currently, measures for controlling soil-suspended dust are limited, primarily relying on long-term strategies such as afforestation and increasing urban green coverage, thus requiring a longer process and sustained investment.

Sand dust transport serves as a significant source of CM in the North China Plain (Zhang et al., 2024). The Ca/Al ratio from 2016 to 2022 (Fig. 3b) shows minimal variation, with annual averages ranging between 1.5 and 2.5, indicating no significant changes in the source regions of sand. The transport trajectories reveal that the predominant pathways for long-distance transport of sand dust originated from Inner Mongolia, passing through Shaanxi and Shanxi provinces. Compared to 2013–2018 (45%), the influence of long-distance transport decreased to 25% during 2019–2022. In contrast, local transport within Henan province and short-distance transport from Shandong province showed a

noticeable increase. These findings suggest that the rebound in CM concentrations during 2019–2022
in Zhengzhou might be responsible for the resuspension of surrounding soil dust.

## 3.4 Long-term trend of particle pH

Are shown in Fig. 4 and Table S4, pH values showed a clearly increasing trend after 2014. From
2013 to 2019, the annual pH increased by 0.11 units from the LS slope [9.15E–04 units from Sen's
slope], reaching a maximum median value of 4.45 (Mean: 4.35) in 2018. Note that the annual average
growth rate of pH values increased to 0.21 units from LS slope [2.93E–03 units from Sen's slope] from
2019 to 2022, with a maximum median value of 4.42 (Mean: 4.51) in 2022. Seasonally, pH values
showed increasing trends in spring, summer, and autumn, and notably increased in winter from 2020
to 2022 (Fig. S9). The increasing trend in pH values observed in this study is similar to the findings in
Beijing (Song et al., 2019; Xie et al., 2020), presumably attributable to the comparable chemical
composition trends and meteorological conditions. In contrast, Shanghai and Hong Kong display
divergent trends (Nah et al., 2023; Zhou et al., 2022). This disparity might be ascribed to the stronger
buffering effect exerted by $NH_3$ and dust in Zhengzhou than marine aerosols ($Na^+/Cl^-$) in these coastal
cities (Shi et al., 2017; Liu et al., 2019). Moreover, the relatively higher temperatures and more
abundant rainfall in Shanghai and Hong Kong could also contribute to the distinct trends observed in
their pH values.
Sensitivity analyses were conducted to explore the dominant factors driving the elevated particle
pH in Zhengzhou by giving a range for one parameter (i.e., $TNH_x$) and average values for other
parameters (i.e., $SO_4^{2-}$, $NO_3^-$, $Na^+$, $Cl^-$, $Ca^{2+}$, $K^+$, $Mg^{2+}$, RH, and T) input into the ISORROPIA-II model.
Are shown in Fig. S10, particle pH increases with the cation concentrations (e.g., $TNH_x$, $K^+$, $Ca^{2+}$,
$Mg^{2+}$, and $Na^+$) and decreases with anions concentrations (e.g., $SO_4^{2-}$ and $NO_3^-$). Additionally, RH
does not significantly affect pH, whereas an increase in T leads to a noticeable decrease in particle pH.

326        Based on the sensitivity analysis curves, the pH values corresponding to a variable in different

years were calculated according to the average values of this variable in different years (Table S6).
The difference in pH values of this variable between two adjacent years was defined as ΔpH which is
illustrated in Fig. 5. According to Equation (2), in addition to $H^+$ concentration, particle pH is primarily
influenced by the dilution effect of ALWC. Moreover, ALWC affects the gas-particle partitioning of
semi-volatile compounds, thereby influencing particle acidity (Zuend et al., 2010; Zuend and Seinfeld,
2012). As shown in Fig.5 and Table S6, only in 2015, 2019, and 2020 did the increases in ALWC
concentration (17.6 μg/m³, 4.1 μg/m³, and 11.6 μg/m³, respectively) lead to pH increases of 0.22, 0.06,
and 0.14 units. This clearly cannot fully explain the significant pH increase in Zhengzhou since 2013.
Notably, since 2013, $H^+$ concentration has shown a decreasing trend. Particularly, $H^+$ concentrations
decreased by $7.6 \times 10^{-6}$, $11.2 \times 10^{-6}$, and $7.2 \times 10^{-6}$ mol/L in 2013, 2015, and 2017, respectively,
leading to pH increases of 0.21, 0.36, and 0.42 units. After 2019, a continuous decline in $H^+$
concentration was observed for three consecutive years, resulting in pH increases of 0.21, 0.13, and
0.2 units in 2020, 2021, and 2022, respectively. These findings indicate that the increase in pH from
2019 to 2022 in Zhengzhou was primarily driven by the reduction in $H^+$ concentration.

341        The concentration of $H^+$ in the aerosol liquid phase is influenced by both chemical composition

and meteorological conditions. To further understand the factors affecting ΔpH, we analyzed the
variations in $PM_{2.5}$ chemical components and meteorological parameters. Results indicate that the
decline in $SO_4^{2-}$ from 2013 to 2018 was the primary cause of the increase in particle pH, as it decreased
$H^+$ and ALWC concentrations (Fig. S11) in aerosol (Ding et al., 2019; Zhang et al., 2021). The average
$SO_4^{2-}$ concentration decreased by 14.6 and 5.3 $\mu g/m^3$, resulting in a pH increase of 0.43 and 0.35 units
from 2013 to 2014 and 2016 to 2017, respectively, which was comparable to an increased rate of 0.3
units in East Asia due to $SO_2$ emission controls since 2016 (Karydis et al., 2021). As another acidic
ion, the decrease in nitrate concentration did not significantly contribute to the pH increases, consistent
with findings from Ding et al. (2019) and Zhang et al. (2021). This is primarily because $NO_3^-$ declined
more slowly compared to sulfate ions and exceeded sulfate concentrations after 2016, under which
conditions that nitrate-rich particles can absorb twice the amount of water that sulfate-rich particles,
leading to an increase in ALWC concentration and inhibiting pH decline (Lin et al., 2020; Xie et al.,
2020). On the other hand, increases in particle pH in 2015 and 2018 were notably influenced by
changes in $TNH_x$ with concentrations increased by 5.5 and 1.3 $\mu g/m^3$, respectively. Increased $TNH_x$
concentrations could react with $SO_4^{2-}$ or $NO_3^-$ and consume a substantial amount of $H^+$, thereby raising
particulate matter pH values (Seinfeld et al., 1998; Zhang et al., 2021). Substantial decreases in T in
2015 (4.2°C), 2017 (4.9°C), and 2018 (2.8°C), favoring $NH_3$ partitioning into the particle phase and
reducing $H^+$ concentrations, drove increases in particle pH (Tao and Murphy, 2019).
During the period from 2020 to 2022, the influence of $SO_4^{2-}$ on particle pH gradually decreased,
with a decrease in concentration from 0.3 to 2.3 $\mu g/m^3$ (Table S6) only bringing about a pH decrease
of 0.03 to 0.14 (Fig. 5). Moreover, a rebound in $SO_4^{2-}$ concentration to 7.9 ± 4.5 $\mu g/m^3$ in 2022 even
resulted in a decrease of 0.11 units in pH instead. On the other hand, TNHx began to show a slight
annual decline (0.9 to 2.2 $\mu g/m^3$), resulting in a significant decrease in pH (0.21–0.35). Consequently,
the increase in pH values was closely related to the rise in $Ca^{2+}$ concentration. $Ca^{2+}$ is less volatile and
competes preferentially with $NH_3$ to neutralize anions such as $SO_4^{2-}$ to form insoluble $CaSO_4$, which
precipitates from the aerosol aqueous phase (Ding et al., 2019; Karydis et al., 2021), thereby reducing
$H^+$ concentrations (Fig. S11) and subsequently lowering particle acidity. Specifically, increases of 0.7
and 0.5 $\mu g/m^3$ in $Ca^{2+}$ concentrations led to pH increases of 0.13 and 0.09 units in 2020 and 2022,
respectively, making $Ca^{2+}$ a primary controlling factor for pH elevation.

## 4 Conclusions

The annual average $PM_{2.5}$ concentration in Zhengzhou decreased from $212.4 \pm 101.5$ $\mu g/m^3$ in
2013 to $59.5 \pm 41.2$ $\mu g/m^3$ in 2022, with the largest reduction in $SO_4^{2-}$. As for CM, their concentrations
notably decreased from 2013 to 2019, because of effective dust control measures, as well as decreased
wind speed and increased relative humidity. However, the proportions of CM in $PM_{2.5}$ have shown
consecutive annual increases. In addition, CM concentrations and their proportions increased
significantly during 2019–2022, which might be responsible for the resuspension of surrounding soil
dust. Correspondingly, the annual pH increased by 0.11 units from 2013 to 2019 mainly due to the
decline in $SO_4^{2-}$, increased $TNH_x$, or decreased temperature. During the period from 2020 to 2022, the
annual average growth rate of pH values increased to 0.21 units from 2019 to 2022, which was
determined by the rise in $Ca^{2+}$ concentration.

## 5 Implication

Control measures implemented by the Chinese government have proven effective in reducing dust, but this study reveals that the crustal materials in $PM_{2.5}$ rebounded after 2019. This phenomenon is not unique to Zhengzhou, as major cities in the North China Plain have also experienced a pronounced upward trend in coarse particles after 2019 (Fig. S12). Thus, crustal materials persist as a substantial constituent of atmospheric aerosols in North China, sustaining elevated particle pH levels. Extensive research has established that heightened particle pH inhibits nitrate reduction in aerosols (Ding et al., 2019; Lin et al., 2020; Wen et al., 2018), particularly significant given nitrate's predominant role in haze formation within this region. Notably, while moderately acidic aerosols demonstrate reduced health impacts, particles with pH < 3 exhibit substantially greater health risks (Shi et al., 2019). Consequently, future environmental management strategies must prioritize real-time assessment of regulatory impacts on particle acidity. This necessitates an integrated approach that simultaneously addresses acidic precursors, alkaline precursors, and crustal material contributions to atmospheric acid chemistry.

## Data availability

All the data presented in this article can be accessed through https://doi.org/10.5281/zenodo.14032007 (Zhang, 2024).

## Supporting Information

Additional data, figures, and tables, some of which are referenced directly within the manuscript, and detailed descriptions of field measurements and samples.

## Author contributions

S.W. designed this study. H.Z. and Z.D. analyzed the data and prepared the manuscript with the contributions of all coauthors. X.L. conducted measurements. R.Z. provided funding acquisition. All authors have read and agreed to the published version of the manuscript.

## Competing interests

The authors declare that they have no conflict of interest.

## Acknowledgment

This work was supported by the National Key Research and Development Program of China (No. 2024YFC3713701), the China Postdoctoral Science Foundation (2023 M733220), the Zhengzhou $PM_{2.5}$ and $O_3$ Collaborative Control and Monitoring Project (20220347 A), and the National Key R&D Program of China No. 2017YFC0212400.

## Funding Sources

This work was supported by the National Key Research and Development Program of China (No. 2024YFC3713701), the China Postdoctoral Science Foundation (2023 M733220), the Zhengzhou $PM_{2.5}$ and $O_3$ Collaborative Control and Monitoring Project (20220347 A), and the National Key Research and Development Program of China (No. 2017YFC0212400).

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

## Figures

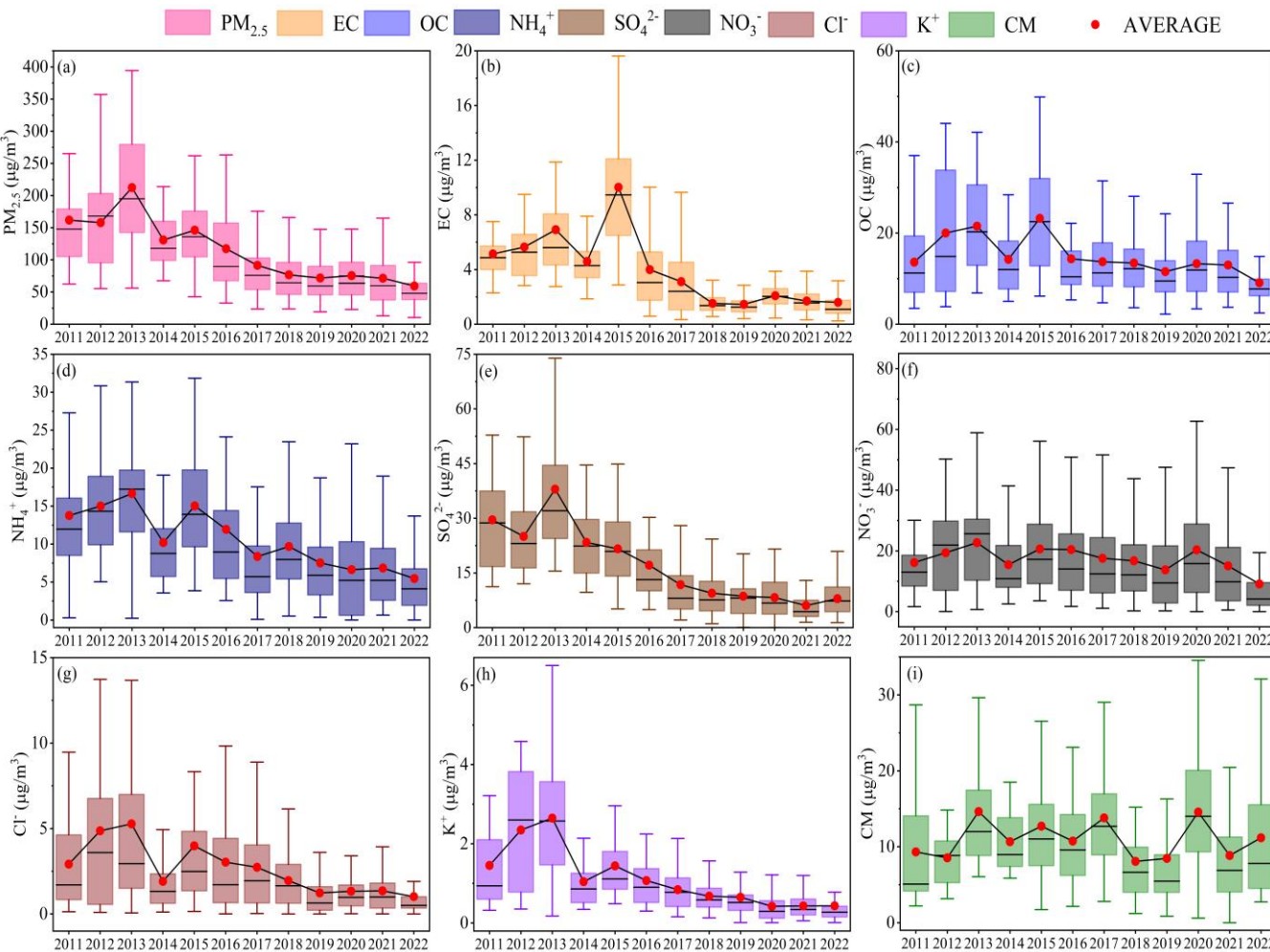

Figure 1. Long-term trends in the concentrations of PM$_{2.5}$ and its chemical components in from 2011 to 2022 in Zhengzhou. Box plots depict annual averages (red dots) and medians (black lines), the top, middle, and bottom lines represent the 75, 50, and 25 percentiles of statistical data, respectively, and the upper and lower whiskers represent the 90 and 10 percentiles of statistical data, respectively.

641

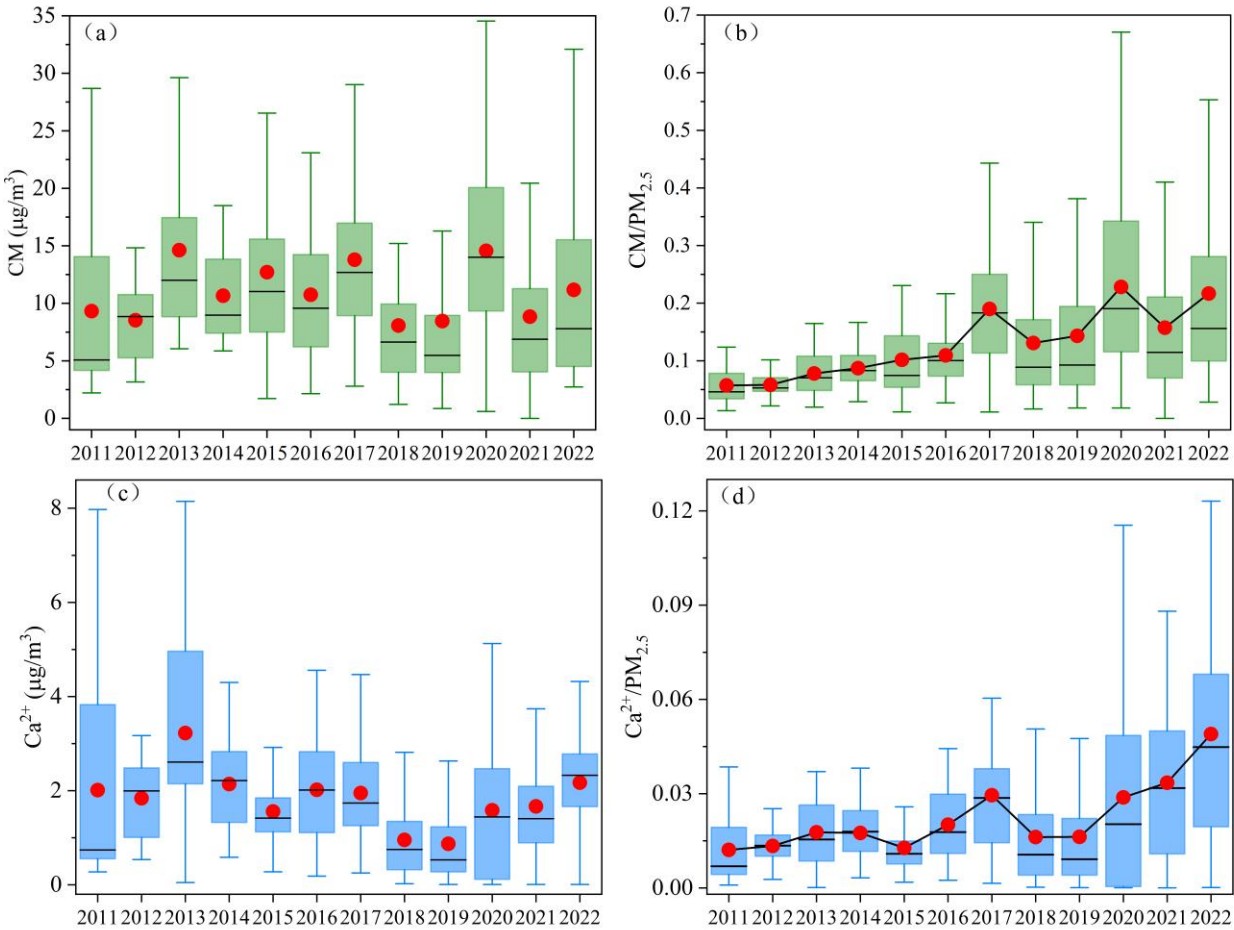

Figure 2. (a) and (c) Long-term trends in CM and $Ca^{2+}$ concentrations in Zhengzhou from 2011 to 2022, respectively. Box plots depict annual averages (red dots) and medians (black lines). (b) and (d) Long-term trends in the proportions of CM and $Ca^{2+}$ in $PM_{2.5}$, respectively.

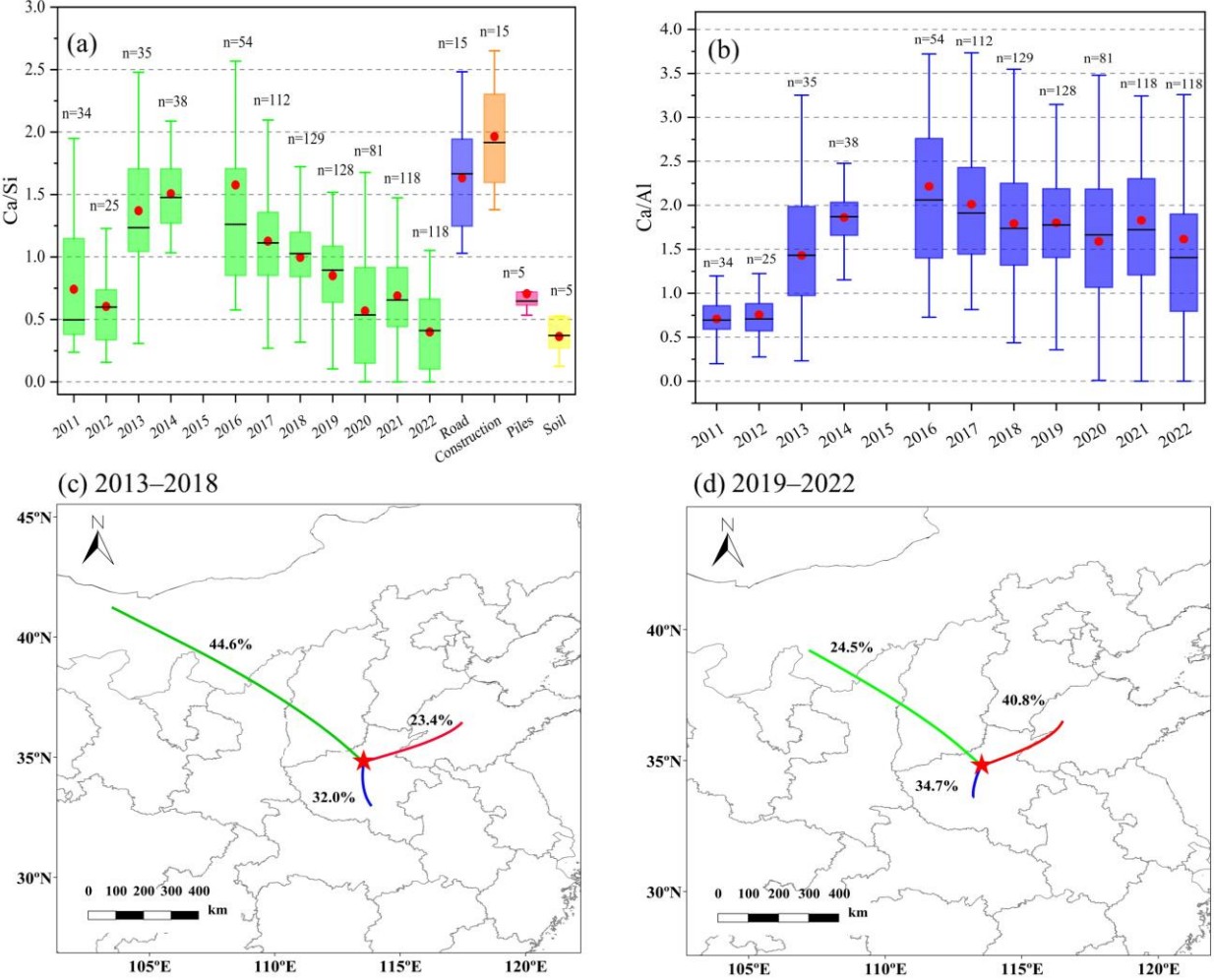

Figure 3. (a) The annual Ca/Si ratios in Zhengzhou from 2011 to 2022 compared with those in various dust sources (specific values and references in Table S5). The red dots and black lines in the box plots represent the annual averages and medians, respectively, with n indicating the sample size. (b) The Ca/Al ratios in Zhengzhou from 2011 to 2022. The red dots and black lines in the box plots represent the annual averages and medians, respectively, with n indicating the sample size. (c) and (d) The transport pathways of CM during 2013–2018 and 2019–2022, respectively.

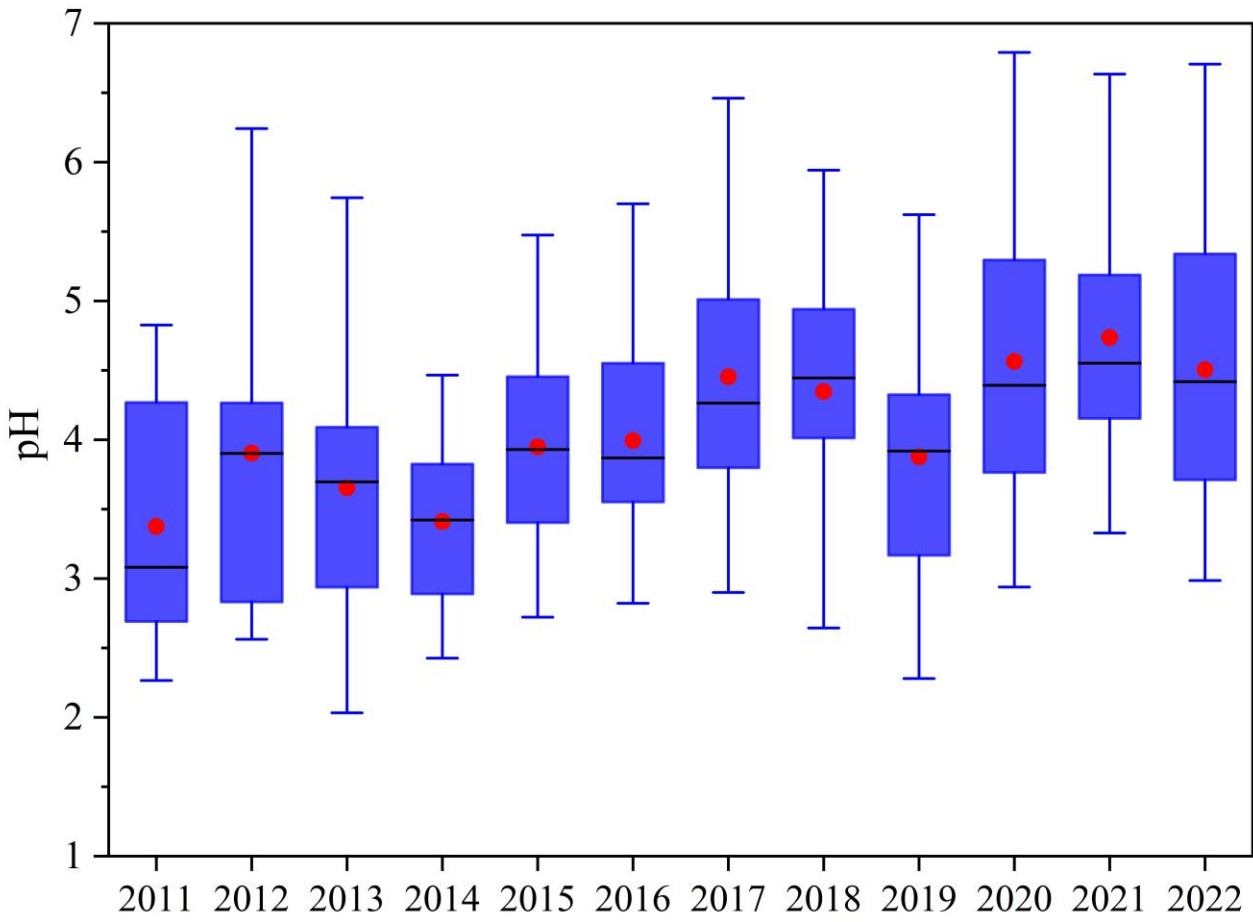


Figure 4. The time series of particle pH in Zhengzhou from 2011 to 2022. In the boxplots, red dots
and black lines represent the annual mean and median values, respectively.

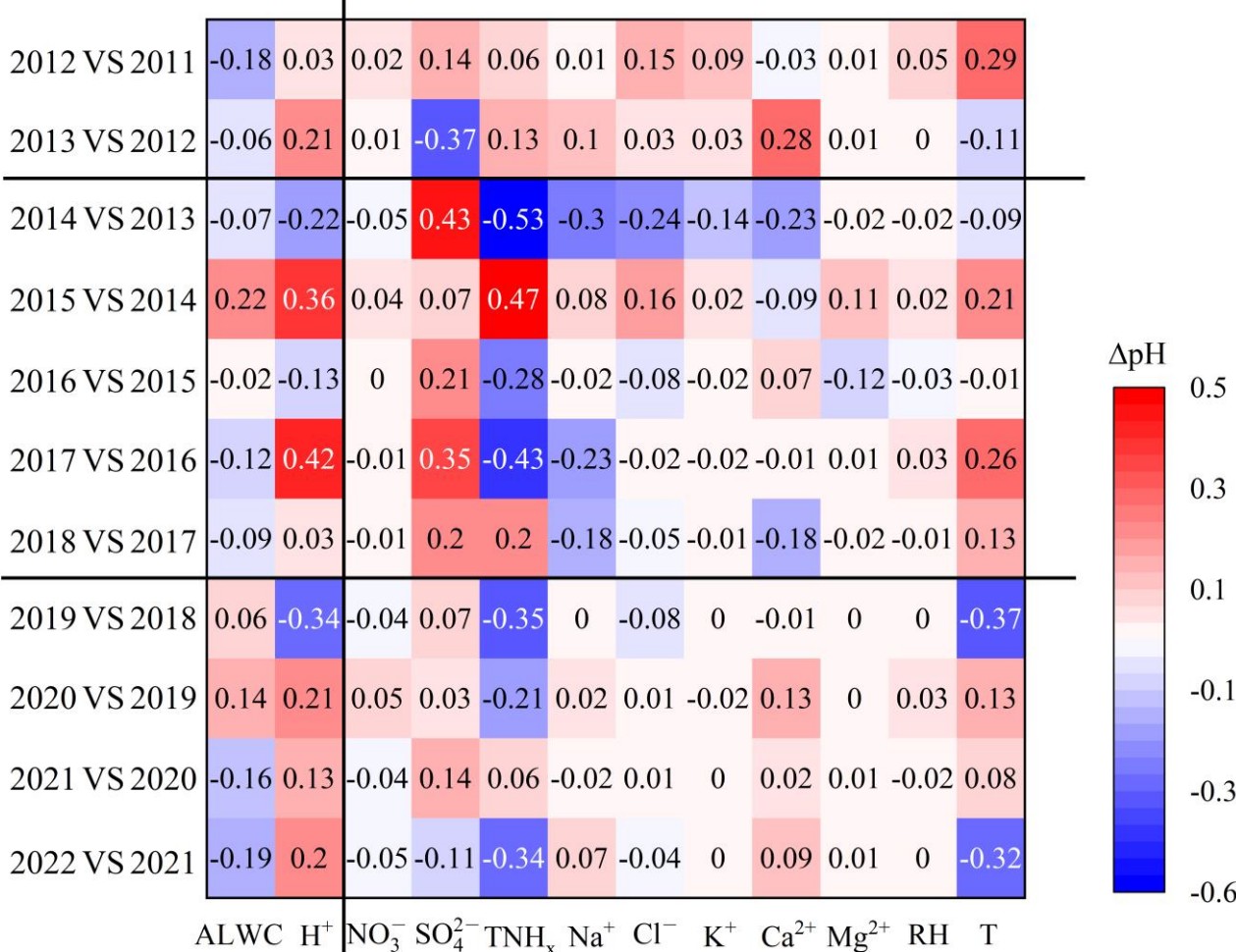


Figure 5. Contribution of each component to the changes in pH (ΔpH) between adjacent years. The difference between component concentrations and meteorological parameters between adjacent years is listed in Table S6.

**Table**
Table 1. Annual average concentrations of $PM_{2.5}$ and its components from 2011 to 2022 in Zhengzhou,
China ($\mu g/m^3$).

| Years | $PM_{2.5}$ | EC | OC | $NO_3^-$ | $SO_4^{2-}$ | $NH_4^+$ | CM | $Ca^{2+}$ |
|---|---|---|---|---|---|---|---|---|
| 2011 | 161.9±81.4 | 5.1±2.1 | 13.6±8.6 | 16.2±11.2 | 29.6±14.3 | 13.8±8.3 | 9.3±7.3 | 2.0±2.2 |
| 2012 | 157.9±71.2 | 5.6±2.5 | 20.0±13.4 | 20.2±13.7 | 25.0±11.2 | 15.0±7.1 | 8.5±3.4 | 1.8±0.8 |
| 2013 | 212.4±101.5 | 6.9±3.8 | 21.5±10.4 | 22.7±13.2 | 38.0±19.9 | 17.1±6.9 | 14.6±8.3 | 3.2±2.1 |
| 2014 | 130.8±48.7 | 4.6±2.0 | 14.2±8.2 | 15.5±10.8 | 23.4±9.3 | 10.2±6.2 | 10.7±4.4 | 2.1±1.0 |
| 2015 | 146.1±61.0 | 10.0±4.7 | 23.2±11.6 | 20.6±14.5 | 21.6±9.8 | 15.7±7.5 | 12.7±6.8 | 1.6±0.7 |
| 2016 | 117.4±73.5 | 4.0±2.8 | 14.4±10.0 | 20.4±18.7 | 17.1±11.3 | 11.9±10.6 | 10.8±5.3 | 2.0±1.1 |
| 2017 | 91.5±61.1 | 3.1±2.5 | 13.7±7.5 | 17.6±15.9 | 11.8±11.6 | 8.4±7.9 | 13.8±6.5 | 2.0±1.0 |
| 2018 | 76.8±41.6 | 1.5±0.7 | 13.4±7.3 | 16.7±13.5 | 9.4±6.0 | 9.7±6.1 | 8.1±5.7 | 1.0±0.8 |
| 2019 | 68.4±34.8 | 1.5±0.8 | 11.5±6.8 | 13.8±13.9 | 8.6±6.4 | 7.5±6.1 | 8.5±7.8 | 0.9±0.9 |
| 2020 | 75.5±31.8 | 2.1±0.9 | 13.3±7.9 | 18.6±14.2 | 8.3±5.6 | 6.7±6.6 | 14.6±7.6 | 1.6±1.4 |
| 2021 | 71.5±45.9 | 1.7±0.9 | 13.0±8.0 | 15.1±15.1 | 6.1±4.5 | 6.8±6.0 | 8.9±7.0 | 1.7±1.2 |
| 2022 | 59.5±41.1 | 1.6±1.5 | 9.1±8.1 | 10.0±14.4 | 7.9±4.5 | 5.5±5.4 | 11.2±8.3 | 2.2±1.1 |

