# Peer review of "Measurement report: Crustal materials play an increasing role in elevating"

_EGUsphere, 2024_

## Author Comment (AC1)

**No.: egusphere-2024-2869**

**Title: Measurement report: Crustal materials play an increasing role in elevating particle pH: Insights from 12-year records in a typical inland city of China.**

**Reviewer #1:**

**General Comments:**

In this article, the authors analyze 12 years of field observation data from Zhengzhou, China, to investigate trends in aerosol composition concentrations and its acidity. This approach is well-aligned with the scope and objectives of the "Measurement Report", making the study highly relevant. However, there are contentious aspects in the discussion and final conclusions of this article. Specifically, whether crustal materials play a dominant role in driving the changes in aerosol acidity remains uncertain. This issue affects several parts of the article, including its main argument and even the title. I believe the authors are capable of making substantial revisions to the article.

Thank you for your careful reading of our paper and valuable comments and suggestions. We believe that we have adequately addressed your comments. To facilitate your review, we used yellow highlights for your comments, green highlights for Reviewer #2, and red color indicating our own corrections in the manuscript.

**Major issues:**

1. Lines 15-17: Compared to the reduction of ammonia emissions, the reductions in acidic precursors such as $SO_2$ and $NO_x$ have been more significant, leading to an overall increase in atmospheric acidity (including aerosols, clouds, and precipitation). Studying the trends in aerosol acidity is highly meaningful, but it is essential to consider the combined impact of the reduction in both acidic and basic precursors.

**Response:** Thanks for your suggestion. This sentence has been modified to: "Particle acidity serves as a key determinant in atmospheric chemical processes. Emerging concerns regarding aerosol acidity trends have been highlighted amid China's sustained initiatives to mitigate emissions of both acidic and alkaline precursors."

2. Lines 19-21: The 12-year observation period and the corresponding years for the PM concentration results do not align. While I understand that the authors began observations in 2011, the abstract should be rephrased to ensure clarity and consistency in the presentation of the time frame.

**Response:** Sorry for the misunderstanding. This sentence has been modified to: "12-year observational data in Zhengzhou reveal that the annual average $PM_{2.5}$ concentration decreased from $162 \pm 81$ μg/m$^3$ in 2011 to $60 \pm 41$ μg/m$^3$ in 2022, with the largest reduction in sulfate (73%). Correspondingly, the annual particle pH increased by 0.10 units from 2011 to 2019."

3. Lines 28-29: According to mainstream forecasts regarding the need for further PM reduction under China's carbon peaking and carbon neutrality policy, future emission reduction strategies will primarily focus on acidic precursors. The expression here needs to be more precise and cautious.

**Response:** Thank you for your comment. This sentence has been modified to: "Therefore, the long-term evolution of particle acidity in North China will require comprehensive consideration of synergistic effects involving acidic precursors, ammonia, and crustal materials."

4. Lines 63-65: There is a lack of logical flow from the previous discussion of acidity changes to the conclusion/summary in this sentence. Additionally, aerosol acidity is unlikely to be neutral or even approach neutrality by nature in general.

**Response:** Sorry for the misunderstanding. This sentence has been modified to: "The atmospheric behavior of ammonium, governed by gas-particle partitioning processes involving ammonia ($NH_3$) as the predominant alkaline gas, demonstrates notable stability in concentration levels, with observational records showing less than 5% interannual variation in $NH_3$ column densities over North China during 2015–2019 (Dong et al., 2023). Under such conditions, the dominant inorganic aerosol component transitions from ammonium sulfate to ammonium nitrate. This compositional shift enhances atmospheric particulate hygroscopicity due to ammonium nitrate's superior water uptake capability, ultimately elevating particle pH levels through aqueous-phase dilution mechanisms (Pinder et al., 2007, 2008; Heald et al., 2012; Weber et al., 2016)."

5. Lines 66-67: This may not be entirely accurate. The relative contribution of sulfate and nitrate does not directly determine aerosol acidity. The authors need to identify the true driving factors behind the pH trend. In my opinion, the main drivers are the $NH_3/NH_4^+$ multiphase buffering, ALWC, and non-volatile cations, rather than other components or temperature. This is because, over the long term, the impact of temperature variations on pH is minimal within the same season.

**Response:** Thank you for your comment. The observed transition in inorganic aerosol composition from ammonium sulfate to ammonium nitrate fundamentally alters aerosol hygroscopicity, as evidenced by deliquescence relative humidity differentials: $NH_4NO_3$ exhibits DRH = 61% versus 80% for $(NH_4)_2SO_4$ at 298K (ISORROPIA II model). This phase shift enhances water uptake by 38–72% (Wexler and Seinfeld, 1991). The consequent dilution effect on hydrogen ion concentration ($[H^+]$) leads to pH increases.

Therefore, this sentence has been modified to: "This compositional shift enhances atmospheric particulate hygroscopicity due to ammonium nitrate's superior water uptake capability, ultimately elevating particle pH levels through aqueous-phase dilution mechanisms (Wexler and Seinfeld, 1991; Pinder et al., 2007, 2008; Heald et al., 2012; Weber et al., 2016). For instance, a significant increase in the nitrate-to-sulfate molar ratio from 2014–2017 in Beijing resulted in the particle pH increasing from 4.4

to 5.4 (Xie et al., 2020)."

6. Lines 85-86: Given the limitations, under what circumstances does it go from 4 to 7?.

**Response:** Thank you for your careful reading of our paper. This sentence has been modified to: "Karydis et al. (2021) framework demonstrated that CM played a critical buffering role in sustaining aerosol pH around 7 across the Middle East arid regions. The model sensitivity tests revealed that under hypothetical dust-free conditions (CM = 0), aerosol acidity would escalate to pH~4 due to $NH_4^+/SO_4^{2-}$ domination."

7. Lines 93-94: After reviewing the research progress, I suggest concluding with a summary that introduces the focus of this article. Specifically, what makes Zhengzhou and other cities different, what issue this study aims to address based on previous research, and what contributions this study makes. This is essential in scientific writing.

**Response:** Thank you for your comment. We have added a description in revised version:

"Zhengzhou presents unique atmospheric chemistry that distinguishes it from other mega-cities in North China. As the capital of China's foremost agricultural province (Henan Province, contributing 18% of national $NH_3$ emissions), Zhengzhou's $PM_{2.5}$ composition combined substantial crustal material (15 ± 3% in $PM_{2.5}$ vs. <10% in Beijing) with exceptional ammonia abundance (Huang et al., 2012; Liu et al., 2018;

Wang et al., 2018). This created distinct particle acidity characteristics, maintaining pH 4.5–6.0 compared to lower pH levels (3.3–5.4) in other cities like Beijing (Ding et al.,2019; Zhang et al., 2021). However, two critical research gaps persist: (1) the long-term evolution of CM under control policies remains unquantified; (2) the role of CM on pH buffer capacity in $NH_3$-enriched environments lacks systematic assessment.

To address these gaps, our study pioneers the first multi-decadal analysis (2011–2022) coupling $PM_{2.5}$ components with thermodynamic modeling through three key innovations: (1) revealing the long-term trends of CM, (2) analyzing the variations of CM sources, and (3) exploring pH trend and its relationship with CM. The resultant findings advance our understanding of urban aerosol acidity chemistry by underscoring the critical role of CM."

8. Line 123: Provide the version of ISORROPIA and the time resolution of the input components.

**Response:** Thank you for your suggestion. This sentence has been modified to: "The particle pH was calculated using the ISORROPIA-II mode (version 2.1, http://isorropia.eas.gatech.edu). The input data (excluding RH ≤ 30%), including $SO_4^{2-}$, $TNO_3$ ($HNO_3$ + $NO_3^-$), $TNH_x$ ($NH_3$+$NH_4^+$), $Ca^{2+}$, $K^+$, $Na^+$, $Mg^{2+}$, $Cl^-$, RH and T, with the temporal resolution aligned with the sampling periods (from 10:00 AM to 9:00 AM the following day)."

9. Line 160: The spring of 2021 saw multiple rare dust storms in the North China Plain, but why was the CM not high in 2021? Additionally, the resolution of the figures in both the main text and SI needs to be improved to at least 300 dpi, as many figures in the SI are unclear.

**Response:** Thank you for your comments.

"The quality assurance protocol excluded temporally discrete dust storm and precipitation periods to prevent contamination of the source analysis of CM and modeling particle pH, given that such events induce non-representative extremes in both crustal element concentrations and pH values, coupled with elevated PM measurement uncertainties." In the revised manuscript, this quality control protocol description has been incorporated into Section 2.1 of the Methodology chapter.

Additionally, the resolution of the figures in both the main text and SI has been adjusted to 600 dpi.

10. Line 170 (Figure S3): It is recommended to use the same y-axis range for consistency. Regarding the drivers of seasonal variations (increase or decrease), I believe the discussion here lacks sufficient rigor, with an incomplete chain of evidence. I suggest that the authors consider separately discussing natural and anthropogenic sources, especially the impact of spring dust storm events on dust levels. In section 3.3, the source-related discussion is also mentioned, and therefore, this part needs to be strengthened.

**Response:** Thank you for your comments. We have replotted the Figure S5.

[Figure]

Figure S5. Trends in the CM concentrations in different seasons from 2011 to 2022.

We agree with your comment that the explanation of the seasonal variation characteristics lacked precise evidence. Therefore, we have revised the descriptions for different seasons and added figures showing the trends of wind speed (WS) and relative humidity (RH) changes across different seasons. However, as mentioned earlier, the data for dust storm events have been removed, and therefore, we cannot distinguish between natural and anthropogenic sources of dust. The revised description is as follows:

"Seasonal trends (Fig. S5) reveal significant declines in CM during spring in 2013–2019 with WS decreasing from 2.2 m/s in 2013 to 1.4 m/s in 2019 (Fig. S6) and stable RH (Fig. S7). Similarly, summer CM reductions in 2013–2019 corresponded with WS declines. These patterns suggest spring-summer CM improvements resulted from the synergistic effects of meteorological changes and dust control policies. Conversely, autumn-winter seasons showed limited CM reductions despite comparable WS decreases in 2013–2019, highlighting the need for enhanced dust emission controls in Zhengzhou during these seasons."

[Figure]

Figure S6 The variation in WS across different seasons from 2011 to 2022.

[Figure]

Figure S7 The variation in RH across different seasons from 2011 to 2022.

11. Lines 239-241: I believe ALWC is an important factor influencing pH, and it should not be solely attributed to sulfate. The authors need to carefully consider this point. Specifically, it should be evaluated whether the continuous decrease in ALWC affects pH, and whether this impact might even outweigh the influence of changes in aerosol composition.

**Response:** Thank you for your comments. We added the description on ALWC:

"According to Equation (2), in addition to $H^+$ concentration, particle pH is primarily influenced by the dilution effect of ALWC. Moreover, ALWC affects the gas-particle partitioning of semi-volatile compounds, thereby influencing particle acidity (Zuend et al., 2010; Zuend and Seinfeld, 2012). As shown in Fig.5 and Table S6, only in 2015, 2019, and 2020 did the increases in ALWC concentration (17.6 μg/m³, 4.1 μg/m³, and 11.6 μg/m³, respectively) lead to pH increases of 0.22, 0.06, and 0.14 units. This clearly cannot fully explain the significant pH increase in Zhengzhou since 2013. Notably, since 2013, $H^+$ concentration has shown a decreasing trend. Particularly, $H^+$ concentrations decreased by $7.6 \times 10^{-6}$, $11.2 \times 10^{-6}$, and $7.2 \times 10^{-6}$ mol/L in 2013, 2015, and 2017, respectively, leading to pH increases of 0.21, 0.36, and 0.42 units. After 2019, a continuous decline in $H^+$ concentration was observed for three consecutive years, resulting in pH increases of 0.21, 0.13, and 0.2 units in 2020, 2021, and 2022, respectively. These findings indicate that the increase in pH from 2019 to 2022 in Zhengzhou was primarily driven by the reduction in $H^+$ concentration.

The concentration of $H^+$ in the aerosol liquid phase is influenced by both chemical composition and meteorological conditions. To further understand the factors affecting ΔpH, we analyzed the variations in $PM_{2.5}$ chemical components and meteorological parameters. Results indicate that the decline in $SO_4^{2-}$ from 2013 to 2018 was the primary cause of the increase in particle pH, as it decreased $H^+$ and ALWC concentrations (Fig. S11) in aerosol (Ding et al., 2019; Zhang et al., 2021)."

[Figure]

Figure 1. Contribution of each component to the changes in pH (ΔpH) between adjacent years. The difference between ALWC, $H^+$, particle component concentrations, and meteorological parameters between adjacent years is listed in Table S6.

Table S6. The difference between component concentrations (μg/m$^3$) and meteorological parameters between adjacent years.

| Years | ALWC | H$^+$(10$^{-6}$) | NO$_3^-$ | SO$_4^{2-}$ | TNH$_X$ | Na$^+$ | Cl$^-$ | K$^+$ | Ca$^{2+}$ | Mg$^{2+}$ | RH(%) | T (℃) |
|---|---|---|---|---|---|---|---|---|---|---|---|---|
| 2012VS2011 | −19.0 | −1.5 | 4.0 | −4.6 | 1.3 | 0.02 | 2.0 | 0.9 | −0.2 | 0.04 | −9.6 | −5.7 |
| 2013VS2012 | −4.6 | −7.6 | 2.6 | 13.0 | 2.1 | 0.2 | 0.4 | 0.3 | 1.4 | 0.1 | −2.6 | 2.1 |
| 2014VS2013 | −4.5 | 7.9 | −7.3 | −14.6 | −6.9 | −0.4 | −3.4 | −1.6 | −1.1 | −0.2 | 6.6 | 2.0 |
| 2015VS2014 | 17.6 | −11.2 | 5.2 | −1.8 | 5.5 | 0.1 | 2.1 | 0.4 | −0.6 | 0.6 | −5.6 | −4.2 |
| 2016VS2015 | −2.3 | 3.0 | −0.2 | −4.5 | −3.7 | −0.03 | −0.1 | −0.4 | 0.5 | −0.7 | 8.0 | 0.3 |
| 2017VS2016 | −10.0 | −7.2 | −2.9 | −5.3 | −3.6 | −0.2 | −0.3 | −0.2 | −0.1 | 0.1 | −6.0 | −4.9 |
| 2018VS2017 | −5.8 | −0.3 | −0.8 | −2.4 | 1.3 | −0.1 | −0.8 | −0.2 | −0.1 | −0.1 | 1.4 | −2.8 |
| 2019VS2018 | 4.1 | 4.8 | −3.0 | −0.8 | −2.2 | −0.04 | −0.7 | −0.03 | −0.1 | −0.01 | −0.1 | 7.3 |
| 2020VS2019 | 11.6 | −3.4 | 4.9 | −0.3 | −0.9 | 0.1 | 0.1 | −0.2 | 0.7 | 0.02 | −6.6 | −2.1 |
| 2021VS2020 | −12.7 | −1.5 | −3.6 | −2.3 | 0.2 | −0.01 | 0.03 | 0.01 | 0.1 | 0.04 | −2.8 | −1.5 |
| 2022VS2021 | −10.6 | −1.5 | −5.1 | 1.9 | −1.4 | 0.03 | −0.3 | 0.01 | 0.5 | 0.04 | 4.7 | 5.8 |

12. Lines 280-282: Where is the data for this section sourced from?

**Response:** This dataset was obtained from the China National Environmental Monitoring Center (CNEMC) and has been referenced in Section 2.1:

"The annual mean $PM_{2.5}$ concentration data for cities in the North China Plain were obtained from the China National Environmental Monitoring Center (CNEMC), available at https://www.cnemc.cn/."

13. Lines 283-286: Although many researchers have done significant work on the impact of ammonia reduction on PM levels, I believe that under China's current policies, future emission reductions will not primarily focus on ammonia. Therefore, the authors need to reconsider whether the focus of this "Measurement Report" should be placed on ammonia. I suggest that the authors revise all statements related to ammonia, including the title.

**Response:** We sincerely thank the reviewer for their thoughtful and constructive feedback. Your point about the non-primary focus on ammonia reduction under current Chinese policies is well taken, and the sentence has been modified to: "
[revised manuscript text omitted]
). The analysis of EC and OC was conducted in two stages. In the first stage, the filter membrane was placed in a quartz heating furnace under a pure helium atmosphere. As the temperature gradually increased to approximately 580°C, OC was volatilized and released. In the second stage, heating continued in a mixed atmosphere of 2% oxygen and 98% helium. When the temperature reached approximately 870°C, EC underwent oxidative decomposition and was released. During the helium flow transmission, OC and EC released at different temperatures were completely oxidized to $CO_2$ in a $MnO_2$ oxidation furnace and subsequently reduced to $CH_4$ for detection by a flame ionization detector (FID). The detection limits for both OC and EC were 0.2 μg/cm². Before each sample analysis, calibration was performed using a standard sucrose solution. Additionally, parallel tests were conducted every ten samples to ensure accuracy.

Water-soluble inorganic ions ($Cl^-$, $NO_3^-$, $SO_4^{2-}$, $Na^+$, $NH_4^+$, $K^+$, $Mg^{2+}$, and $Ca^{2+}$) were measured using ion chromatography (ICS-90 and ICS-900 models, Dionex, USA). Half of the $PM_{2.5}$ filter was cut into pieces and ultrasonically extracted with 20 mL of Milli-Q water for 30 min, followed by filtering through a 0.45 mm polytetrafluoroethylene syringe filter before analysis. The cation concentrations were determined by an IonPacASll-HC4 mm anion separation column and an IonPacAGll-HC4 mm guard column, with 20 mM methane sulfonate as an eluent at 0.8 mL/min. The anions were measured by an IonPacCS12A cation separation column and an IonPacCG12A guard column, with a solution of 8.0 mM $Na_2CO_3$ + 1.0 mM $NaHCO_3$ as an eluent at 1.0 mL/min. The regression coefficients ($R^2$) of the calibration curves were over 0.9996 for all ions, except $NH_4^+$ (0.9988), which showed a quadratic response.

Elements were analyzed using a wavelength dispersive X-ray fluorescence spectrometer (S8 TIGER, Bruker, Germany) to determine concentrations of Fe, Na, Mg, Al, Si, Cl, K, Ca, V, Ni, Cu, Zn, Cr, Mn, Co, Cd, Ga, As, Se, Sr, Sn, Sb, Ba, and Pb (Tremper et al., 2018), which has been approved by the United States Environmental Protection Agency (Chow and Watson, 1994). The spectrometer was equipped with an X-ray tube featuring close coupling among the tube, sample, and detector, ensuring high efficiency and optimal excitation of elements within the sample. Before analysis, the instrument was calibrated using a series of high-quality, self-prepared standards. Calibration procedures were conducted following established methods (Chow and Watson, 1994). To assess potential contamination and ensure data quality, blank filters were routinely analyzed alongside each batch of samples.

**Figures**

[Figure]

Figure S1.Sampling site in Zhengzhou, China. © 2019 National Geomatics Center of China. All
rights reserved.

.

[Figure]

Figure S2. Verification of the NH₃ Method.

[Figure]

Figure S3. Trends in the proportions of chemical components in PM₂.₅ from 2011 to 2022.

[Figure]

Figure S4. Trends in the meteorological parameters from 2011 to 2022.

[Figure]

Figure S5. Trends in the CM concentrations in different seasons from 2011 to 2022.

[Figure]

Figure S6. The variation in WS across different seasons from 2011 to 2022.

[Figure]

Figure S7. The variation in RH across different seasons from 2011 to 2022.

[Figure]

Figure S8. Trends in the concentrations of crustal elements and their proportions in PM$_{2.5}$ from 2011 to 2022.

[Figure]

Figure S4. Trends in the particle pH in different seasons from 2011 to 2022.

Figure S5. Sensitivity analysis of input parameters to particle pH. The dashed line represents the
average of the observational data from 2011 to 2022.

[Figure]

Figure S6. Trends in aerosol liquid water content (ALWC) and H$^+$ concentrations from 2011 to 2022

[Figure]

Figure S7. Trends in the annual average concentrations of PM$_{2.5}$, PM$_{10}$, and PM$_{10-2.5}$ in provincial
capitals in the North China Plain.

 **Tables**

 Table S1. Information on sampling date and numbers.

| Years | Sampling date | The effective number of samples |
|---|---|---|
| 2011 | April 7–20
July 1–31
October 28–December 2
December 11–November 23 | 188 |
| 2012 | February 25–26
April 21–May 6
July 22–August 2
October 17–November 1
December 8–25 | 140 |
| 2013 | February 25–March 6
April 1–May 1
June 5–July 30
September 20–October 13
December 2–18 | 184 |
| 2014 | April 1–May 5
June 18–July 20
October 7–24
December 30–31 | 180 |
| 2015 | January 1–15
April 1–20
July 1–20
October 9–24 | 248 |
| 2016 | January 6–22
April 8–30
July 9–31
October 1–20
December 29–31 | 252 |
| 2017 | January 1–20
April 18–May 4
July 1–26
October 14–December 31 | 480 |
| 2018 | January 1–31
April 1–30
July 1–31
October 9–December 31 | 600 |
| 2019 | January 1–31
April 1–30
July 1–31
September 2–October 31
November 12–30
December 21–31 | 592 |

| Year | Dates | Count |
|---|---|---|
| 2020 | January 1–20
June 5–July 31
October 6–November 13
December 15–31 | 332 |
| 2021 | January 1–31
March 16–April 30
July 1–August 8
October 17–December 31 | 540 |
| 2022 | January 1–4
April 1–May 3
July 1–August 11
September 5–October 11
December 10–31 | 492 |
| Total | | 4228 |

Table S2. The method detection limit (MDL) and measurement uncertainties (Unc) of individual components

| | MDLs ($\mu g/m^3$) | Unc (%) |
|---|---|---|
| EC | 0.1 | 13.1 |
| OC | 0.1 | 9.8 |
| $Na^+$ | 0.005 | 9.6 |
| $NH_4^+$ | 0.011 | 10.1 |
| $K^+$ | 0.006 | 9.5 |
| $Mg^{2+}$ | 0.002 | 9.3 |
| $Ca^{2+}$ | 0.017 | 8.8 |
| $F^-$ | 0.001 | 8.2 |
| $Cl^-$ | 0.001 | 9.3 |
| $NO_3^-$ | 0.015 | 10.1 |
| $SO_4^{2-}$ | 0.031 | 9.9 |
| Na | 0.003 | 10.9 |
| Mg | 0.002 | 10.6 |
| Al | 0.004 | 9.2 |
| Si | 0.005 | 9.3 |
| Cl | 0.008 | 9.5 |
| K | 0.005 | 9.4 |
| Ca | 0.01 | 9.4 |
| V | 0.008 | 57.9 |
| Ni | 0.006 | 96.6 |
| Cr | 0.02 | 24.7 |
| Mn | 0.02 | 16.8 |
| Fe | 0.03 | 9.3 |
| Co | 0.009 | 79.6 |
| Cu | 0.005 | 5.8 |
| Zn | 0.003 | 8.4 |
| Ga | 0.005 | 84.7 |
| As | 0.008 | 27.4 |

| Se | 0.006 | 25.7 |
| Sr | 0.006 | 22.8 |
| Cd | 0.03 | 68.6 |
| Sn | 0.02 | 42.0 |
| Sb | 0.02 | 73.6 |
| Ba | 0.02 | 15.6 |
| Pb | 0.02 | 13.4 |

Table S3. Control measures for dust implemented by Henan Province and Zhengzhou government

| Release time | Policies | Regulatory focus |
|---|---|---|
| 2013.9 | Regulations on Reducing Pollutant Emissions in Henan Province | Road, Construction |
| 2014.8 | Temporary Regulations on Dust Control Management at Construction Sites in Henan Province | Construction |
| 2016.7 | Implementation Plan for Controlling Dust Pollution in Henan Province | Road, Construction |
| 2018.2 | Regulations on the Prevention and Control of Atmospheric Pollution in Henan Province | Road, Construction, Piles |
| 2019.4 | Special Action Plan for Fine Management of Dust Pollution Prevention and Control at Construction Sites in Zhengzhou City, 2019 | Construction |
| 2019.8 | Enhanced Action Plan for Intensive Dust Control at Construction Sites in 2019 | Construction |
| 2021.1 | Special Governance Plan for Key Project Dust Pollution in Zhengzhou | Road, Construction, Piles |

Table S4. Analysis of the inter - annual trends of CM and $Ca^{2+}$ concentrations and pH during different periods using multiple methods.

| | 2011–2013 | | | 2013–2019 | | | 2019–2022 | | |
|---|---|---|---|---|---|---|---|---|---|
| | CM | $Ca^{2+}$ | pH | CM | $Ca^{2+}$ | pH | CM | $Ca^{2+}$ | pH |
| MK-$Z$ | 3.01 | 2.70 | 1.41 | −9.74 | −13.62 | 3.00 | 2.48 | 8.21 | 5.12 |
| MK-$p$ | 0.003 | 0.007 | 0.159 | <2.2 E–16 | <2.2E–16 | 0.003 | 0.013 | 2.20E–16 | 2.99E–07 |
| Sen's slope | 0.082 | 0.023 | 7.10E–03 | −0.015 | −4.14E–03 | 9.15E–04 | 5.80E–03 | 5.42E–03 | 2.93E–03 |
| LS slope | 2.65 | 0.61 | / | −0.81 | −0.32 | 0.11 | 0.24 | 0.40 | 0.21 |

** MK-$Z$ and MK-$p$ represent the trend (Z) and significance ($p$) calculated by the Mann - Kendall method using daily data, respectively; Sen's slope represents the Sen slope using daily data; LS slope represents the Least - Squares slope using annual data. All the above calculations were performed using the R language (R version 4.0.2).

[revised manuscript text omitted]

---

## Author Comment (AC4)

**No.: egusphere-2024-2869**

**Title: Measurement report: Crustal materials play an increasing role in elevating particle pH: Insights from 12-year records in a typical inland city of China.**

**Reviewer #2:**

**General Comments:**

Zhang et al. analyzed the particle pH collected in a Chinese inland city. This analysis is based on 4228 filter samples collected from four seasons in 2011 – 2022. As a measurement report, the study fits into the scope of Atmospheric Chemistry and Physics. The authors claim that the evolving particle pH was driven by the interplay of declines in $SO_4^{2-}$, increases in $TNH_x$, and rises in $Ca^{2+}$ concentrations over time. The manuscript can be considered for publication once the comments below are addressed.

Thank you for your careful reading of our paper and valuable comments and suggestions. We believe that we have adequately addressed your comments. To facilitate your review, we used green highlights for your comments, yellow highlights for Reviewer #1, and red color indicating our own corrections in the manuscript.

**Major Comment:**

1. lines 111-112 and 120–121: It is unclear whether Ti was measured or not.

**Response:** Sorry for the mistake. We did not measure the element Ti. The sentence has been corrected:

"Elements were analyzed using a wavelength dispersive X-ray fluorescence spectrometer (S8 TIGER,

Bruker, Germany) to determine concentrations of Fe, Na, Mg, Al, Si, Cl, K, Ca, V, Ni, Cu, Zn, Cr, Mn, Co, Cd, Ga, As, Se, Sr, Sn, Sb, Ba, and Pb (Tremper et al., 2018)."

2. Lines 135 – 136: The linear regression proposed by Wei et al. (2023) is based on the dataset collected in the wintertime but not from 2013 to 2020. This needs clarification. Since the regression was based on wintertime data, it is questionable if the relationship is still valid for data in seasons other than winter.

**Response:** Thanks for your comment. We have added a validation of the simulated $NH_3$: "To validate the applicability of Equation 4 for annual $NH_3$ estimation and pH simulation in Zhengzhou, this study utilized both observed $NH_3$ data (from a Thermo Scientific URG-9000D ambient ion monitor, USA) and calculated $NH_3$ values derived from Equation 4 at the same monitoring site throughout 2022, inputting them into the thermodynamic model for pH simulation. As shown in Figure S2, pH values calculated from observed and simulated $NH_3$ exhibit good agreement ($r = 0.97$, $P < 0.01$). Furthermore, $NH_3$ concentrations modeled by ISORROPIA demonstrate a significant correlation ($r = 0.95$, $P < 0.01$) with that simulated $NH_3$ by Equation 4. These results collectively demonstrate the reliability of the $NH_3$ estimation method in this study."

[Figure]

Figure S2. Verification of the NH$_3$ Method.

3. Section 2.2.3: Why was 1000 m chosen as the height in the HYSPLIT simulations? If 1000 m was already well above the boundary layer height, how do the simulated back trajectory simulations represent the ground-level measurements? In addition, how was the optimum number of clusters chosen by the authors? What clustering technique was used in the analysis? What data was used as the meteorological input?

**Response:** Thanks for your comments.

While the surface elevation of Zhengzhou is approximately 100 m above sea level (ASL), setting the height at 1000 m ASL takes into account the minimum altitude needed to traverse the average elevation of the Taihang Mountains (ranging from 1000 to 1500 m ASL). This ensures that the simulated trajectory paths over this topographical barrier are physically realistic.

The Angle Distance algorithm was used to cluster air mass trajectories, enabling the identification of dominant air mass directions and potential pollution sources affecting the study site during different periods. The optimal number of clusters was determined by evaluating the spatial variance (SPVAR)

of each trajectory from the cluster mean and the total spatial variance (TSV). The final classification was selected just before the second rapid increase in TSV. The underlying principle is that TSV initially rises sharply during clustering, then increases gradually; however, once the number of clusters reaches a certain threshold, TSV surges again, indicating that the merged clusters are highly dissimilar, marking the end of the classification process. The classification results correspond to the different air mass categories before this final merging step. The mean trajectories of these clusters represent the primary airflow patterns at the target site during the analysis period (Wang et al., 2009).

The HYSPLIT simulations utilized meteorological input data from the Global Data Assimilation System (GDAS) with 3D wind vectors, temperature, relative humidity, geopotential height, surface pressure, and boundary layer diagnostics.

In the revised version, these descriptions have been added in Section 2.2.3:

"Backward trajectories were calculated using the mixed-particle Lagrangian integrated trajectory method (HYSPLIT, https:// www.ready.noaa.gov/HYSPLIT_traj.php). Meteorological input data were from the Global Data Assimilation System (GDAS) with 3D wind vectors, temperature, relative humidity, geopotential height, surface pressure, and boundary layer diagnostics. 24-h backward trajectories were simulated for air masses above 1000 m above ground level in Zhengzhou. While the surface elevation of Zhengzhou is approximately 100 m above sea level (ASL), setting the height at 1000 m ASL takes into account the minimum altitude needed to traverse the average elevation of the Taihang Mountains (ranging from 1000 to 1500 m ASL). This ensures that the simulated trajectory paths over this topographical barrier are physically realistic.

The Angle Distance algorithm was used to cluster air mass trajectories, enabling the identification of dominant air mass directions and potential pollution sources affecting the study site during different periods. The optimal number of clusters was determined by evaluating the spatial variance (SPVAR) of each trajectory from the cluster mean and the total spatial variance (TSV). The final classification was selected just before the second rapid increase in TSV. The underlying principle is that TSV initially rises sharply during clustering, then increases gradually; however, once the number of clusters reaches a certain threshold, TSV surges again, indicating that the merged clusters are highly dissimilar, marking the end of the classification process. The classification results correspond to the different air mass categories before this final merging step. The mean trajectories of these clusters represent the primary airflow patterns at the target site during the analysis period (Wang et al., 2009). Subsequently, trajectories from two periods, 2013–2018 and 2019–2022, were clustered separately to analyze the variations between the two policy implementation periods."

4. Sections 3.1 and 3.2: How did the author estimate the decreases in the mass concentrations? Which year was chosen as the reference year? A proper analysis using Mann-Kendall and Sen's slope should be carried out here and in other places associated with trend analysis.

**Response:** Thank you for your comment. In the revised manuscript, the Mann - Kendall, Sen's slope, and Least - Squares slope methods were comprehensively used to analyze the inter - annual change trends. Please refer to Table S4 for details.

Table S4. Analysis of the inter - annual trends of CM and $Ca^{2+}$ concentrations and pH during different periods using multiple methods.

| | 2011–2013 | | | 2013–2019 | | | 2019–2022 | | |
|---|---|---|---|---|---|---|---|---|---|
| | CM | $Ca^{2+}$ | pH | CM | $Ca^{2+}$ | pH | CM | $Ca^{2+}$ | pH |
| MK-Z | 3.01 | 2.70 | 1.41 | –9.74 | –13.62 | 3.00 | 2.48 | 8.21 | 5.12 |
| MK-$p$ | 0.003 | 0.007 | 0.159 | <2.2 E–16 | <2.2E–16 | 0.003 | 0.013 | 2.20E–16 | 2.99E–07 |
| Sen's slope | 0.082 | 0.023 | 7.10E–03 | –0.015 | –4.14E–03 | 9.15E–04 | 5.80E–03 | 5.42E–03 | 2.93E–03 |
| LS slope | 2.65 | 0.61 | / | –0.81 | –0.32 | 0.11 | 0.24 | 0.40 | 0.21 |

** MK-Z and MK-$p$ represent the trend (Z) and significance ($p$) calculated by the Mann - Kendall method using daily data, respectively; Sen's slope represents the Sen slope using daily data; LS slope represents the Least - Squares slope using annual data. All the above calculations were performed using the R language (R version 4.0.2).

5. Please briefly describe the Air Pollution Prevention and Control Action Plan and Three-Year Action Plan in the main text. In general, readers outside China have no idea about these policies.

**Response:** Thank you for your comments. We have added a description of the policies: "Over the past twelve years, the Chinese government implemented the Air Pollution Prevention and Control Action Plans (2013–2018) and the Three-Year Action Plan (2018–2020). The Air Pollution Prevention and Control Action Plan focused on reducing $PM_{2.5}$ concentrations in key regions and aiming to cut $PM_{2.5}$ levels by 10–25% in priority areas over five years. To achieve these goals, it adopted several measures. In terms of industrial restructuring, it mandated the elimination of a large amount of outdated production capacity in industries such as iron/steel and cement to optimize the industrial structure and reduce high-pollution production. For emission standards, it set strict requirements for multiple industrial sectors, especially coal-fired power plants, and gradually introduced ultra-low emission requirements to control pollutants like $SO_2$, $NO_x$, and PM. Regarding energy transition, it promoted a shift from coal to cleaner energy sources, including capping coal consumption in certain regions and restricting the construction of small-scale coal-fired boilers. Subsequently, the Three-Year Action Plan was carried out to continue improving air quality with a broader scope of regions under control, further reducing pollutant emissions and enhancing the overall air quality index. The measures included enhanced transportation controls, such as introducing stricter vehicle emission standards (like National VI standards for vehicles) and establishing diesel truck exclusion zones in many cities to reduce emissions from the transportation sector. It also adopted precision governance through grid-based environmental supervision, dividing areas into small grids for more accurate and efficient monitoring of pollution sources. Additionally, it strengthened the legal and institutional framework by revising relevant laws, such as the Air Pollution Prevention and Control Law, to strengthen legal penalties for environmental violations and implementing an environmental tax system to encourage enterprises to reduce emissions. Correspondingly, the annual average concentration of $PM_{2.5}$ in Zhengzhou decreased from $212 \pm 102$ μg/m$^3$ in 2013 to $60 \pm 41$ μg/m$^3$ in 2022, representing a reduction of approximately 72%."

6. Please explain why Zhengzhou is representative of a typical inland city in North China in the main text. There is no background information about aerosol research conducted in Zhengzhou and how they compare with those findings in other typical Chinese megacities, such as Beijing or Shanghai.

**Response:** Thanks for your comment. We have added a summary section:

"Zhengzhou presents unique atmospheric chemistry that distinguishes it from other mega-cities in North China. As the capital of China's foremost agricultural province (Henan Province, contributing 18% of national NH$_3$ emissions), Zhengzhou's $PM_{2.5}$ composition combined substantial crustal material ($15 \pm 3\%$ in $PM_{2.5}$ vs. <10% in Beijing) with exceptional ammonia abundance (Huang et al., 2012; Liu et al., 2018; Wang et al., 2018). This created distinct particle acidity characteristics, maintaining pH 4.5–6.0 compared to lower pH levels (3.3–5.4) in other cities like Beijing (Ding et al.,2019; Zhang et al., 2021). However, two critical research gaps persist: (1) the long-term evolution of CM under control policies remains unquantified; (2) the role of CM on pH buffer capacity in NH$_3$-enriched environments lacks systematic assessment.

To address these gaps, our study pioneers the first multi-decadal analysis (2011–2022) coupling $PM_{2.5}$ components with thermodynamic modeling through three key innovations: (1) revealing the long-term trends of CM, (2) analyzing the variations of CM sources, and (3) exploring pH trend and its relationship with CM. The resultant findings advance our understanding of urban aerosol acidity chemistry by underscoring the critical role of CM."

**Minor Comment:**

1. Figure S1: Please label the highways, coal-fired power plants, and gas-fired power plants on the map.

**Response:** Thanks for your comment. We have updated Figure S1:

[Figure]

Figure S1.Sampling site in Zhengzhou, China. © 2019 National Geomatics Center of China. All rights reserved.

2. Section 2.1: What are the uncertainties in the measured concentrations for individual components?

**Response:** Thanks for your comment. The method detection limits and measurement uncertainties are summarized in Table S2.

Table S2 The method detection limit (MDL) and measurement uncertainties (Unc) of individual components

|  | MDLs ($\mu g/m^3$) | Unc (%) |
|---|---|---|
| EC | 0.1 | 13.1 |
| OC | 0.1 | 9.8 |
| $Na^+$ | 0.005 | 9.6 |
| $NH_4^+$ | 0.011 | 10.1 |
| $K^+$ | 0.006 | 9.5 |
| $Mg^{2+}$ | 0.002 | 9.3 |
| $Ca^{2+}$ | 0.017 | 8.8 |
| $F^-$ | 0.001 | 8.2 |
| $Cl^-$ | 0.001 | 9.3 |
| $NO_3^-$ | 0.015 | 10.1 |
| $SO_4^{2-}$ | 0.031 | 9.9 |
| Na | 0.003 | 10.9 |
| Mg | 0.002 | 10.6 |
| Al | 0.004 | 9.2 |
| Si | 0.005 | 9.3 |
| Cl | 0.008 | 9.5 |
| K | 0.005 | 9.4 |
| Ca | 0.01 | 9.4 |
| V | 0.008 | 57.9 |
| Ni | 0.006 | 96.6 |
| Cr | 0.02 | 24.7 |
| Mn | 0.02 | 16.8 |
| Fe | 0.03 | 9.3 |
| Co | 0.009 | 79.6 |
| Cu | 0.005 | 5.8 |
| Zn | 0.003 | 8.4 |
| Ga | 0.005 | 84.7 |
| As | 0.008 | 27.4 |
| Se | 0.006 | 25.7 |
| Sr | 0.006 | 22.8 |
| Cd | 0.03 | 68.6 |
| Sn | 0.02 | 42.0 |
| Sb | 0.02 | 73.6 |
| Ba | 0.02 | 15.6 |
| Pb | 0.02 | 13.4 |

3. Lines 114 – 115: Apart from just citing references here, please provide the details about analytical methods and quality control in the supplement.

**Response:** Thanks for your comments. We have added detailed analytical methods and quality control in both the Manuscript and Supplement material:

**Manuscript**

"Blank filters were also routinely analyzed with each batch of samples to detect sample contamination and provide quality assurance on the elemental concentrations. Detailed analytical methods and quality control are described in the supplement (Text S1). The method detection limits and measurement uncertainties are summarized in Table S2."

**Supplement material**

**"Text S1 Instruments and Measurements**

Samples were collected using a high-volume sampler (TE-6070D, Tisch, USA) and air particulate samplers (TH-16A, Tianhong, China) from April 2011 to December 2022. Two quartz filters and two Teflon filters were used daily from 10:00 AM to 9:00 AM the next day, resulting in a total of 5848 samples. After excluding abnormal data due to instrument malfunctions, 4228 valid samples were obtained. Detailed information on the samples is provided in Table S1.

Organic carbon (OC) and elemental carbon (EC) were analyzed using a carbon analyzer (Model 5L, Sunset Laboratory, USA). The analysis of EC and OC was conducted in two stages. In the first stage, the filter membrane was placed in a quartz heating furnace under a pure helium atmosphere. As the temperature gradually increased to approximately 580°C, OC was volatilized and released. In the second stage, heating continued in a mixed atmosphere of 2% oxygen and 98% helium. When the temperature reached approximately 870°C, EC underwent oxidative decomposition and was released.

During the helium flow transmission, OC and EC released at different temperatures were completely oxidized to $CO_2$ in a $MnO_2$ oxidation furnace and subsequently reduced to $CH_4$ for detection by a flame ionization detector (FID). The detection limits for both OC and EC were 0.2 μg/cm². Before each sample analysis, calibration was performed using a standard sucrose solution. Additionally, parallel tests were conducted every ten samples to ensure accuracy.

Water-soluble inorganic ions ($Cl^-$, $NO_3^-$, $SO_4^{2-}$, $Na^+$, $NH_4^+$, $K^+$, $Mg^{2+}$, and $Ca^{2+}$) were measured using ion chromatography (ICS-90 and ICS-900 models, Dionex, USA). Half of the $PM_{2.5}$ filter was cut into pieces and ultrasonically extracted with 20 mL of Milli-Q water for 30 min, followed by filtering through a 0.45 mm polytetrafluoroethylene syringe filter before analysis. The cation concentrations were determined by an IonPacASll-HC4 mm anion separation column and an IonPacAGll-HC4 mm guard column, with 20 mM methane sulfonate as an eluent at 0.8 mL/min. The anions were measured by an IonPacCS12A cation separation column and an IonPacCG12A guard column, with a solution of 8.0 mM $Na_2CO_3$ + 1.0 mM $NaHCO_3$ as an eluent at 1.0 mL/min. The regression coefficients ($R^2$) of the calibration curves were over 0.9996 for all ions, except $NH_4^+$ (0.9988), which showed a quadratic response.

Elements were analyzed using a wavelength dispersive X-ray fluorescence spectrometer (S8 TIGER, Bruker, Germany) to determine concentrations of Fe, Na, Mg, Al, Si, Cl, K, Ca, V, Ni, Cu, Zn, Cr, Mn, Co, Cd, Ga, As, Se, Sr, Sn, Sb, Ba, and Pb (Tremper et al., 2018), which has been approved by the United States Environmental Protection Agency (Chow and Watson, 1994). The spectrometer was equipped with an X-ray tube featuring close coupling among the tube, sample, and detector, ensuring high efficiency and optimal excitation of elements within the sample. Before analysis, the instrument was calibrated using a series of high-quality, self-prepared standards. Calibration procedures were conducted following established methods (Chow and Watson, 1994). To assess potential contamination and ensure data quality, blank filters were routinely analyzed alongside each batch of samples."

4. Section 2.2.2: What was the set activity coefficient of H in the model?

**Response:** With ISORROPIA, $\gamma H^+$ and $\gamma OH^-$ are assumed to be equal to unity, whereas the activity coefficients for the other ionic pairs (e.g., $H^+$–$Cl^-$) are calculated by Kusik-Meissner method (Fountoukis and Nenes, 2007).

*Fountoukis, C. and Nenes, A.: ISORROPIA II: a computationally efficient thermodynamic equilibrium model for $K^+$–$Ca^{2+}$–$Mg^{2+}$–$NH_4^+$–$Na^+$–$SO_4^{2-}$–$NO_3^-$–$Cl^-$—$H_2O$ aerosols, Atmos. Chem. Phys., 7, 4639–4659, https://doi.org/10.5194/acp-7-4639-2007, 2007.*

5. Line 172: There is no data showing that the WS was higher in spring and summer compared to autumn and winter.

**Response:** According to the comments of Reviewer 1, this part has been revised to:

"Seasonal trends (Fig. S5) reveal significant declines in CM during spring in 2013–2019 with WS decreasing from 2.2 m/s in 2013 to 1.4 m/s in 2019 (Fig. S6) and stable RH (Fig. S7). Similarly, summer CM reductions in 2013–2019 corresponded with WS declines. These patterns suggest spring-summer CM improvements resulted from the synergistic effects of meteorological changes and dust control policies. Conversely, autumn-winter seasons showed limited CM reductions despite comparable WS decreases in 2013–2019, highlighting the need for enhanced dust emission controls in

Zhengzhou during these seasons."

[Figure]

Figure S6 The variation in WS across different seasons from 2011 to 2022.

[Figure]

6. Figure S4: Is there any data for $Ca^{2+}$ in 2015?

**Response:** Thank you for your comments. The relevant data for $Ca^{2+}$ has been presented in Figure 2 of the main text.

7. Lines 229 – 231: Please provide a discussion about why the pH trend is similar to those in Beijing but different from those in Shanghai and Hong Kong.

**Response:** Thanks for your suggestion. We have added a discussion: "The increasing trend in pH values observed in this study is similar to the findings in Beijing (Song et al., 2019; Xie et al., 2020), presumably attributable to the comparable chemical composition trends and meteorological conditions. In contrast, Shanghai and Hong Kong display divergent trends (Nah et al., 2023; Zhou et al., 2022). This disparity might be ascribed to the stronger buffering effect exerted by $NH_3$ and dust in Zhengzhou than marine aerosols ($Na^+$/$Cl^-$) in these coastal cities (Shi et al., 2017; Liu et al., 2019). Moreover, the relatively higher temperatures and more abundant rainfall in Shanghai and Hong Kong could also contribute to the distinct trends observed in their pH values."

8. Figure 5: How did the authors estimate the contribution of a component to the changes in pH?

**Response:** First, we conducted a sensitivity analysis using all the observational data from 2011 to 2022. A given range for a variable (i.e., TNHx) along with the corresponding average values of other parameters (i.e., $SO_4^{2-}$, $NO_3^-$, $Na^+$, $Cl^-$, $Ca^{2+}$, $K^+$, $Mg^{2+}$, RH, and T) was input into ISORROPIA-II to investigate the sensitivity of particle pH to this variable (i.e., TNHx), and the results are presented in

Fig. S10 in the Supplement. Based on the sensitivity analysis curves, the pH values corresponding to a variable in different years were calculated according to the average values of this variable in different years (Table S6). The difference in pH values of this variable between two adjacent years was defined as ΔpH. The above description has also been added to the main text:

"Sensitivity analyses were conducted to explore the dominant factors driving the elevated particle pH in Zhengzhou by giving a range for one parameter (i.e., TNHx) and average values for other parameters (i.e., $SO_4^{2-}$, $NO_3^-$, $Na^+$, $Cl^-$, $Ca^{2+}$, $K^+$, $Mg^{2+}$, RH, and T) input into the ISORROPIA-II model."

"Based on the sensitivity analysis curves, the pH values corresponding to a variable in different years were calculated according to the average values of this variable in different years (Table S6). The difference in pH values of this variable between two adjacent years was defined as ΔpH which is illustrated in Fig. 5"

9. Where is the figure or table associated with Lines 256 – 266?

**Response:** Sorry for the misunderstanding. This sentence has been modified to: "During the period from 2020 to 2022, the influence of $SO_4^{2-}$ on particle pH gradually decreased, with a decrease in concentration from 0.3 to 2.3 μg/m$^3$ (Table S6) only bringing about a pH decrease of 0.03 to 0.14 (Fig. 5).

**Technical Comment:**

1. Line 17: It is unclear what is "particle pH response"

**Response:** Thank you for your careful reading of our paper. We have implemented the following revisions: "Emerging concerns regarding aerosol acidity trends have been highlighted amid China's sustained initiatives to mitigate emissions of both acidic and alkaline precursors, especially in North

China, which is significantly affected by dust aerosol."

2. Line 19: It should be "12-year".

**Response:** Done!

3. Graphical abstract: Please use a different color for the pH. Apparently, the color light blue is too bright for visualization.

**Response:** Done!

[Figure]

4. Line 57: Please use a word different than "research hotspot".

**Response:** OK. We have revised "research hotspot" to "research focus".

5. Line 64: Please use a different term instead of "fostered a persistent belief".

**Response:** This sentence has been deleted referring to Reviewer 1.

6. Eq 4: It should be "$NH_4^+$".

**Response:** Done!

7. Figures in main text and SI: the labels for x- and y- axis and annotations are too small to read. All figures need to be modified for better visualization.

**Response:** Done!

8. Lines 173 and 174: more effective than what?

**Response:** Sorry for the misunderstanding. This sentence has been deleted referring to Reviewer 1.

9. Line 246: $NO_3^-$ can be used to replace nitrate ions.

**Response:** Done!

10. Line 252: $SO_4^{2-}/NO_3^-$ can means the ratio between $SO_4^{2-}$ and $NO_3^-$. I assume the authors use "/" to represent "or" here. If so, please use the word "or"

**Response:** Done!

11. The format of "TNH$_x$" should be consistent throughout the manuscript.

**Response:** Done!

12. Some color choices in figures need to be reconsidered. Certain annotations in color are not color-friendly for visualization.

**Response:** Thanks for your suggestion. The colors of the figures in both the main text and supplement have been adjusted to enhance their readability.

**Manuscript**

[revised manuscript text omitted]
). The analysis of EC and OC was conducted in two stages. In the first stage, the filter membrane was placed in a quartz heating furnace under a pure helium atmosphere. As the temperature gradually increased to approximately 580°C, OC was volatilized and released. In the second stage, heating continued in a mixed atmosphere of 2% oxygen and 98% helium. When the temperature reached approximately 870°C, EC underwent oxidative decomposition and was released. During the helium flow transmission, OC and EC released at different temperatures were completely oxidized to $CO_2$ in a $MnO_2$ oxidation furnace and subsequently reduced to $CH_4$ for detection by a flame ionization detector (FID). The detection limits for both OC and EC were 0.2 μg/cm². Before each sample analysis, calibration was performed using a standard sucrose solution. Additionally, parallel tests were conducted every ten samples to ensure accuracy.

Water-soluble inorganic ions ($Cl^-$, $NO_3^-$, $SO_4^{2-}$, $Na^+$, $NH_4^+$, $K^+$, $Mg^{2+}$, and $Ca^{2+}$) were measured using ion chromatography (ICS-90 and ICS-900 models, Dionex, USA). Half of the $PM_{2.5}$ filter was cut into pieces and ultrasonically extracted with 20 mL of Milli-Q water for 30 min, followed by filtering through a 0.45 mm polytetrafluoroethylene syringe filter before analysis. The cation concentrations were determined by an IonPacASll-HC4 mm anion separation column and an IonPacAGll-HC4 mm guard column, with 20 mM methane sulfonate as an eluent at 0.8 mL/min. The anions were measured by an IonPacCS12A cation separation column and an IonPacCG12A guard column, with a solution of 8.0 mM $Na_2CO_3$ + 1.0 mM $NaHCO_3$ as an eluent at 1.0 mL/min. The regression coefficients ($R^2$) of the calibration curves were over 0.9996 for all ions, except $NH_4^+$ (0.9988), which showed a quadratic response.

Elements were analyzed using a wavelength dispersive X-ray fluorescence spectrometer (S8 TIGER, Bruker, Germany) to determine concentrations of Fe, Na, Mg, Al, Si, Cl, K, Ca, V, Ni, Cu, Zn, Cr, Mn, Co, Cd, Ga, As, Se, Sr, Sn, Sb, Ba, and Pb (Tremper et al., 2018), which has been approved by the United States Environmental Protection Agency (Chow and Watson, 1994). The spectrometer was equipped with an X-ray tube featuring close coupling among the tube, sample, and detector, ensuring high efficiency and optimal excitation of elements within the sample. Before analysis, the instrument was calibrated using a series of high-quality, self-prepared standards. Calibration procedures were conducted following established methods (Chow and Watson, 1994). To assess potential contamination and ensure data quality, blank filters were routinely analyzed alongside each batch of samples.

**Figures**

[Figure]

Figure S1.Sampling site in Zhengzhou, China. © 2019 National Geomatics Center of China. All
rights reserved.
.

[Figure]

Figure S2. Verification of the NH₃ Method.

Figure S3. Trends in the proportions of chemical components in PM₂.₅ from 2011 to 2022.

[Figure]

Figure S4. Trends in the meteorological parameters from 2011 to 2022.

[Figure]

Figure S5. Trends in the CM concentrations in different seasons from 2011 to 2022.

[Figure]

Figure S6. The variation in WS across different seasons from 2011 to 2022.

[Figure]

Figure S7. The variation in RH across different seasons from 2011 to 2022.

[Figure]

Figure S8. Trends in the concentrations of crustal elements and their proportions in PM$_{2.5}$ from 2011 to 2022.

[Figure]

Figure S9. Trends in the particle pH in different seasons from 2011 to 2022.

Figure S10. Sensitivity analysis of input parameters to particle pH. The dashed line represents the
average of the observational data from 2011 to 2022.

[Figure]

Figure S11. Trends in aerosol liquid water content (ALWC) and H$^+$ concentrations from 2011 to 2022

[Figure]

Figure S12. Trends in the annual average concentrations of PM$_{2.5}$, PM$_{10}$, and PM$_{10-2.5}$ in provincial
capitals in the North China Plain.

**Tables**

Table S1. Information on sampling date and numbers.

| Years | Sampling date | The effective number of samples |
|---|---|---|
| 2011 | April 7–20
July 1–31
October 28–December 2
December 11–November 23 | 188 |
| 2012 | February 25–26
April 21–May 6
July 22–August 2
October 17–November 1
December 8–25 | 140 |
| 2013 | February 25–March 6
April 1–May 1
June 5–July 30
September 20–October 13
December 2–18 | 184 |
| 2014 | April 1–May 5
June 18–July 20
October 7–24
December 30–31 | 180 |
| 2015 | January 1–15
April 1–20
July 1–20
October 9–24 | 248 |
| 2016 | January 6–22
April 8–30
July 9–31
October 1–20
December 29–31 | 252 |
| 2017 | January 1–20
April 18–May 4
July 1–26
October 14–December 31 | 480 |
| 2018 | January 1–31
April 1–30
July 1–31
October 9–December 31 | 600 |

| 2019 | January 1–31
April 1–30
July 1–31
September 2–October 31
November 12–30
December 21–31 | 592 |
| --- | --- | --- |
| 2020 | January 1–20
June 5–July 31
October 6–November 13
December 15–31 | 332 |
| 2021 | January 1–31
March 16–April 30
July 1–August 8
October 17–December 31 | 540 |
| 2022 | January 1–4
April 1–May 3
July 1–August 11
September 5–October 11
December 10–31 | 492 |
| Total | | 4228 |

Table S2. The method detection limit (MDL) and measurement uncertainties (Unc) of individual
components

| | MDLs (µg/m³) | Unc (%) |
| --- | --- | --- |
| EC | 0.1 | 13.1 |
| OC | 0.1 | 9.8 |
| $Na^+$ | 0.005 | 9.6 |
| $NH_4^+$ | 0.011 | 10.1 |
| $K^+$ | 0.006 | 9.5 |
| $Mg^{2+}$ | 0.002 | 9.3 |
| $Ca^{2+}$ | 0.017 | 8.8 |
| $F^-$ | 0.001 | 8.2 |
| $Cl^-$ | 0.001 | 9.3 |
| $NO_3^-$ | 0.015 | 10.1 |
| $SO_4^{2-}$ | 0.031 | 9.9 |
| Na | 0.003 | 10.9 |
| Mg | 0.002 | 10.6 |
| Al | 0.004 | 9.2 |
| Si | 0.005 | 9.3 |

| | | |
|---|---|---|
| Cl | 0.008 | 9.5 |
| K | 0.005 | 9.4 |
| Ca | 0.01 | 9.4 |
| V | 0.008 | 57.9 |
| Ni | 0.006 | 96.6 |
| Cr | 0.02 | 24.7 |
| Mn | 0.02 | 16.8 |
| Fe | 0.03 | 9.3 |
| Co | 0.009 | 79.6 |
| Cu | 0.005 | 5.8 |
| Zn | 0.003 | 8.4 |
| Ga | 0.005 | 84.7 |
| As | 0.008 | 27.4 |
| Se | 0.006 | 25.7 |
| Sr | 0.006 | 22.8 |
| Cd | 0.03 | 68.6 |
| Sn | 0.02 | 42.0 |
| Sb | 0.02 | 73.6 |
| Ba | 0.02 | 15.6 |
| Pb | 0.02 | 13.4 |

Table S3. Control measures for dust implemented by Henan Province and Zhengzhou government

| Release time | Policies | Regulatory focus |
|---|---|---|
| 2013.9 | Regulations on Reducing Pollutant Emissions in Henan Province | Road, Construction |
| 2014.8 | Temporary Regulations on Dust Control Management at Construction Sites in Henan Province | Construction |
| 2016.7 | Implementation Plan for Controlling Dust Pollution in Henan Province | Road, Construction |
| 2018.2 | Regulations on the Prevention and Control of Atmospheric Pollution in Henan Province | Road, Construction, Piles |
| 2019.4 | Special Action Plan for Fine Management of Dust Pollution Prevention and Control at Construction Sites in Zhengzhou City, 2019 | Construction |

| | | |
|---|---|---|
| 2019.8 | Enhanced Action Plan for Intensive Dust Control at Construction Sites in 2019 | Construction |
| 2021.1 | Special Governance Plan for Key Project Dust Pollution in Zhengzhou | Road, Construction, Piles |

Table S4. Analysis of the inter - annual trends of CM and $Ca^{2+}$ concentrations and pH during different periods using multiple methods.

| | 2011–2013 | | | 2013–2019 | | | 2019–2022 | | |
| --- | --- | --- | --- | --- | --- | --- | --- | --- | --- |
| | CM | $Ca^{2+}$ | pH | CM | $Ca^{2+}$ | pH | CM | $Ca^{2+}$ | pH |
| MK-$Z$ | 3.01 | 2.70 | 1.41 | –9.74 | –13.62 | 3.00 | 2.48 | 8.21 | 5.12 |
| MK-$p$ | 0.003 | 0.007 | 0.159 | <2.2 E–16 | <2.2E–16 | 0.003 | 0.013 | 2.20E–16 | 2.99E–07 |
| Sen's slope | 0.082 | 0.023 | 7.10E–03 | –0.015 | –4.14E–03 | 9.15E–04 | 5.80E–03 | 5.42E–03 | 2.93E–03 |
| LS slope | 2.65 | 0.61 | / | –0.81 | –0.32 | 0.11 | 0.24 | 0.40 | 0.21 |

** MK-$Z$ and MK-$p$ represent the trend (Z) and significance ($p$) calculated by the Mann - Kendall method using daily data, respectively; Sen's slope represents the Sen slope using daily data; LS slope represents the Least - Squares slope using annual data. All the above calculations were performed using the R language (R version 4.0.2).

 Table S5. The ratios of Ca/Si in the source spectrum of different dust sources in China

| Dust source | City | Ca/Si | Reference |
|---|---|---|---|
| Road dust | Xi'an | 2.04 | http://www.klacp.ac.cn/ |
| | Yinchuan | 2.48 | wgPMzypfypk/ycy/2017 |
| | Lanzhou | 1.67 | 06/t20170610_375562.ht |
| | Beijing | 1.25 | ml |
| | Tianjin | 1.03 | |
| | Baoding | 1.16 | |
| | Shijiazhuang | 1.98 | |
| | Handan | 1.83 | |
| | Shenyang | 1.81 | |
| | Changsha | 1.92 | |
| | Chongqing | 1.38 | |
| | Chengdu | 1.17 | |
| | Kunming | 1.94 | |
| | Taiyuan | 1.55 | |
| | Nanjing | 1.28 | |
| Construction dust | Xi'an | 1.69 | http://www.klacp.ac.cn/ |
| | Yinchuan | 1.84 | wgPMzypfypk/ycy/2017 |
| | Lanzhou | 2.33 | 06/t20170610_375562.ht |
| | Beijing | 2.65 | ml |
| | Tianjin | 1.46 | |
| | Baoding | 1.58 | |
| | Shijiazhuang | 1.38 | |
| | Handan | 1.86 | |
| | Shenyang | 1.92 | |
| | Changsha | 2.30 | |
| | Chongqing | 2.52 | |
| | Chengdu | 2.15 | |
| | Kunming | 1.60 | |
| | Taiyuan | 1.92 | |
| | Nanjing | 2.26 | |
| Piles dust | Xi'an | 0.72 | (Yang, 2016) |
| | Tianjin | 0.57 | (Zhang et al., 2018) |
| | Taiyuan | 0.61 | (Bi et al., 2007) |
| | Jinan | 1.01 | (Bi et al., 2007) |
| | / | 0.65 | http://www.nkspap.com: 9091/Index.aspx |
| Soil dust | Nanchang | 0.37 | (Xu et al., 2019) |
| | Xi'an | 0.27 | (Yang, 2016) |
| | Jincheng | 0.13 | (Wang et al., 2016) |
| | Wuhan | 0.52 | (Gong and Luo, 201 |

| | | | 8) |
|---|---|---|---|
| / | | 0.53 | http://www.nkspap.com: 9091/Index.aspx |

Table S6. The difference between component concentrations ($\mu g/m^3$) and meteorological parameters between adjacent years.

| Years | ALWC | $H^+(10^{-6})$ | $NO_3^-$ | $SO_4^{2-}$ | $TNH_X$ | $Na^+$ | $Cl^-$ | $K^+$ | $Ca^{2+}$ | $Mg^{2+}$ | RH(%) | T (°C) |
|---|---|---|---|---|---|---|---|---|---|---|---|---|
| 2012VS2011 | -19.0 | -1.5 | 4.0 | -4.6 | 1.3 | 0.02 | 2.0 | 0.9 | -0.2 | 0.04 | -9.6 | -5.7 |
| 2013VS2012 | -4.6 | -7.6 | 2.6 | 13.0 | 2.1 | 0.2 | 0.4 | 0.3 | 1.4 | 0.1 | -2.6 | 2.1 |
| 2014VS2013 | -4.5 | 7.9 | -7.3 | -14.6 | -6.9 | -0.4 | -3.4 | -1.6 | -1.1 | -0.2 | 6.6 | 2.0 |
| 2015VS2014 | 17.6 | -11.2 | 5.2 | -1.8 | 5.5 | 0.1 | 2.1 | 0.4 | -0.6 | 0.6 | -5.6 | -4.2 |
| 2016VS2015 | -2.3 | 3.0 | -0.2 | -4.5 | -3.7 | -0.03 | -0.1 | -0.4 | 0.5 | -0.7 | 8.0 | 0.3 |
| 2017VS2016 | -10.0 | -7.2 | -2.9 | -5.3 | -3.6 | -0.2 | -0.3 | -0.2 | -0.1 | 0.1 | -6.0 | -4.9 |
| 2018VS2017 | -5.8 | -0.3 | -0.8 | -2.4 | 1.3 | -0.1 | -0.8 | -0.2 | -0.1 | -0.1 | 1.4 | -2.8 |
| 2019VS2018 | 4.1 | 4.8 | -3.0 | -0.8 | -2.2 | -0.04 | -0.7 | -0.03 | -0.1 | -0.01 | -0.1 | 7.3 |
| 2020VS2019 | 11.6 | -3.4 | 4.9 | -0.3 | -0.9 | 0.1 | 0.1 | -0.2 | 0.7 | 0.02 | -6.6 | -2.1 |
| 2021VS2020 | -12.7 | -1.5 | -3.6 | -2.3 | 0.2 | -0.01 | 0.03 | 0.01 | 0.1 | 0.04 | -2.8 | -1.5 |
| 2022VS2021 | -10.6 | -1.5 | -5.1 | 1.9 | -1.4 | 0.03 | -0.3 | 0.01 | 0.5 | 0.04 | 4.7 | 5.8 |

---

## Author Response (AR2)

**No.: egusphere-2024-2869**

**Title: Measurement report: Crustal materials play an increasing role in elevating particle pH: Insights from 12-year records in a typical inland city of China.**

**Reviewer #2:**

**General Comments:**

The authors have addressed most of my comments. However, there are still a few places that require clarification before the manuscript can be accepted.

Thank you for your careful reading of our paper and valuable comments and suggestions. We believe that we have adequately addressed your comments. To facilitate your review, we used green highlights for your comments, and red color indicating our own corrections in the manuscript.

**Major Comment:**

1. Section 2.2.3: The author stated that the choice of 1000 m ASL as the receptor height was based on the elevation of the Taihang Mountains (1000 – 1500 m ASL). I am not convinced that using 100 m ASL as the receptor height in the HYSPLIT model will lead to unrealistic results. In fact, using 100 m ASL does not influence the movement of an air mass; it can still have a trajectory above 1000 m.

To support the selection of 1000 m ASL, the authors must show consistent HYSPLIT results, irrespective of whether a receptor height of 100 meters or 1000 meters ASL is used during a chosen period for a case study. Otherwise, I recommend that the authors redo the analysis using a receptor height of 100 meters, as this height is more representative of conditions within the boundary layer due

to its proximity to the surface.

**Response:** Thanks for your comment. Upon carefully revisiting the HYSPLIT model documentation and relevant literature, we recognize that selecting a receptor height of 100 m ASL is indeed more appropriate for studying boundary-layer transport processes, as it better represents near-surface pollutant dynamics. In response to your suggestion, we have re-simulated all trajectories using 100 m ASL as the receptor height and updated the methodology, results, and figures in the revised manuscript:

**Section 2.2.3**

"24-hour backward trajectories were simulated for air masses arriving at 100 m above ground level in Zhengzhou, a receptor height aligned with the city's average elevation (~100 m above sea level) to capture near-surface pollutant transport dynamics within the boundary layer."

**Section 3.3**

"The transport trajectories (Fig. 3c and 3d) reveal that a marked decline in the contribution of long-distance sand dust transport originating from Inner Mongolia (via Shaanxi and Shanxi provinces) from 13.9% during 2013–2018 to 7.2% in 2019–2022. In contrast, local transport within Henan province and short-distance transport from Shandong province exhibited contrasting increases. These findings suggest that the rebound in CM concentrations during 2019–2022 in Zhengzhou might be responsible for the resuspension of surrounding soil dust."

[Figure]

Figure 3. (a) The annual Ca/Si ratios in Zhengzhou from 2011 to 2022 compared with those in various dust sources (specific values and references in Table S5). The red dots and black lines in the box plots represent the annual averages and medians, respectively, with n indicating the sample size. (b) The Ca/Al ratios in Zhengzhou from 2011 to 2022. The red dots and black lines in the box plots represent the annual averages and medians, respectively, with n indicating the sample size. (c) and (d) The transport pathways of CM during 2013–2018 and 2019–2022, respectively.

**Minor Comment:**

1. Figure S2: Could the authors clarify why there are more data points in Fig S2 (b) compared with Fig S2 (a)?

**Response:** We sincerely apologize for the lack of clarity in the original figure. The difference in data points between Figures S2(a) and S2(b) arises because Figure S2(a) validation of pH estimates derived

from 2022 observational data vs. simulated $NH_3$ concentrations using Equation 4, while Figure S2(b)

compares $NH_3$ concentrations derived from 12-year (2011–2022) ISORROPIA-II simulations against

Equation 4 calculations.

To resolve this ambiguity, we have revised the figure caption to explicitly state the temporal parameters

and calculation methods. The updated caption now reads:

[Figure]

Figure S2. (a) Validation of pH estimates derived from observed vs. simulated $NH_3$ concentrations in
2022. Red line: linear regression fit. (b) Cross-validation of $NH_3$ concentrations calculated by the
ISORROPIA thermodynamic model and Equation 4 (2011–2022). Red line: linear regression fit.

2. Lines 190 – 201: The statement is too descriptive and long to follow. The authors need to include

a plot showing the SPVAR and TSV as a function of cluster number in the SI.

**Response:** Thanks for your comment. We have made the following revisions to this section:

"Trajectories from 2013–2018 and 2019–2022 were independently clustered via the Angle

Distance algorithm to compare policy-driven variations (Wang et al., 2009). The optimal cluster

number (three, Figure S3) was determined by tracking total spatial variance (TSV), with classification

finalized at the inflection point preceding the second TSV surge."

[Figure]

Figure S3. Evolution of total spatial variance (TSV) and optimal cluster number determination for backward trajectories under two policy phases: (a) 2013–2018; (b) 2019–2022.

3. Line 207: What are the priority areas?

**Response:** Sorry for the misunderstand. This sentence has been modified to: "The 2013–2018 policy prioritized end-of-pipe controls in power generation and heavy industries, mandating ≥10% $PM_{10}$ reduction nationwide and region-specific $PM_{2.5}$ reduction targets (25% for Beijing-Tianjin-Hebei, 20% for Yangtze River Delta, and 15% for Pearl River Delta)."

4. Lines 204 – 224: The information is very descriptive and reads like a report. It needs to be properly shortened to improve the readability.

**Response:** Thanks for your comments. We have modified this section:

"Over the past twelve years, the Chinese government has implemented two major policies to mitigate air pollution: the Air Pollution Prevention and Control Action Plans (2013–2018) and the Three-Year Action Plan (2018–2020), with key targets and measures detailed in Tables S3 and S4. The 2013–2018 policy prioritized end-of-pipe controls in power generation and heavy industries,

mandating $\geq 10\%$ $PM_{10}$ reduction nationwide and region-specific $PM_{2.5}$ reduction targets (25% for Beijing-Tianjin-Hebei, 20% for Yangtze River Delta, and 15% for Pearl River Delta). Subsequently, the 2018–2020 campaign shifted toward structural reforms and multi-pollutant synergistic governance, enforcing $\geq 15\%$ nationwide $SO_2$/$NO_x$ emission cuts and $\geq 18\%$ $PM_{2.5}$ reduction in non-compliant cities relative to 2015 levels."

Table S3. Major emission reduction measures were implemented during the Air Pollution Prevention and Control Action Plan (2013–2018) and the Three-Year Action Plan (2018–2020)

| | Air Pollution Prevention and Control Action Plan | Three-Year Action Plan |
|---|---|---|
| Industrial Restructuring | Elimination of a large amount of outdated production capacity in industries. Cement to optimize the industrial structure Reduce high-pollution production | Continued phase-out of outdated production capacity (e.g., steel, cement) Strengthen ultra-low emission retrofitting in sectors such as steel and coking |
| Energy Transition | Capping coal consumption in certain regions Restricting the construction of small-scale coal-fired power plants | Deepen regional coal consumption control Expand clean heating coverage in rural areas |
| Promote clean fuels in the residential sector | Ultra-low emission retrofitting in the power sector Comprehensive retrofitting of coal-fired boilers | Continue raising emission standards in the power sector Introduce ultra-low emission requirements for non-power sectors (e.g., steel, coking) |
| Mobile Source Control | Initial elimination of high-emission (yellow-label) vehicles | Full implementation of China VI emission standards Set up low-emission zones for diesel trucks |

| | Promotion of China V emission standards | Promote the replacement of vehicles with new energy vehicles |
| --- | --- | --- |
| Coordinated Control of Multiple Pollutants | Focus on $SO_2$, $NO_x$, and $PM_{2.5}$ | Incorporate VOCs (Volatile Organic Compounds) and $NH_3$ (ammonia) into joint control

Strengthen control over industrial solvent use and agricultural emissions |
| Innovative Governance Models | Regional joint prevention and control (key regions such as Beijing–Tianjin–Hebei) | Grid-based and precise supervision (micro-zoning and dynamic management) |

Table S4. The Comparison of Key Indicators Between the Air Pollution Prevention and Control Action Plan (2013–2018) and the Three-Year Action Plan (2018–2020).

| | Air Pollution Prevention and Control Action Plan | Three-Year Action Plan |
|---|---|---|
| $PM_{2.5}$ | Beijing-Tianjin-Hebei region: ≥25% reduction
Yangtze River Delta region: ≥20% reduction
Pearl River Delta region: ≥15% reduction | $PM_{2.5}$ concentrations to be reduced by more than 18%
(vs. 2015) |
| $PM_{10}$ | National $PM_{10}$ concentration to decrease by ≥10% | No explicit targets |
| $SO_2/NO_x$ | No explicit targets | National $SO_2$ and $NO_x$ emissions to be reduced by ≥15%
(vs. 2015) |
| Number of good air quality days | "Annual improvement" in good air days | >80% annually in prefecture-level and above cities |
| Number of heavily polluted days | No explicit targets | 25% reduction in heavy pollution days (vs. 2015) |

5. The response to "What was the set activity coefficient of H in the model?" needs to be included in the section of the method.

**Response:** Done!

"The concentrations of hydrogen ions in air ($H_{air}^{+}$) and ALWC were derived from the $Na^{+}$-$K^{+}$-$Ca^{2+}$-$Mg^{2+}$-$NH_4^{+}$-$SO_4^{2-}$-$NO_3^{-}$-$Cl^{-}$-$H_2O$ equilibrium composition system. Activity coefficients for $H^{+}$ and $OH^{-}$ were fixed at unity, while other ion pairs (e.g., $H^{+}$-$Cl^{-}$) employed the Kusik-Meissner parameterization for ionic activity calculations (Fountoukis and Nenes, 2007)."

6. Lines 314 – 315: Could the authors include references to support the statement "the comparable chemical composition trends and meteorological conditions"?

**Response:** Thank you for your comments. We have added relevant literature citations to support this viewpoint:

"The increasing trend in pH values observed in this study is similar to the findings in Beijing (Song et al., 2019; Xie et al., 2020), presumably attributable to the comparable chemical composition trends and meteorological conditions (Liu et al., 2017; Wang et al., 2020; Xu et al., 2015)."

Liu, M., Song, Y., Zhou, T., Xu, Z., Yan, C., Zheng, M., Wu, Z., Hu, M., Wu, Y., and Zhu, T.: Fine particle pH during severe haze episodes in northern China, Geophys. Res. Lett., 44, 5213–5221, https://doi.org/10.1002/2017GL073210, 2017.

Wang, S.; Wang, L.; Li, Y.; Wang, C.; Wang, W.; Yin, S.; Zhang, R.: Effect of ammonia on fine-particle pH in agricultural regions of China: comparison between urban and rural sites, Atmos. Chem. Phys., 20, 2719–2734, https://doi.org/10.5194/acp-20-2719-2020, 2020.

Wang, J.; Gao, J.; Che, F.; Wang, Y.; Lin, P.; Zhang, Y.: Dramatic changes in aerosol composition

during the 2016–2020 heating seasons in Beijing–Tianjin–Hebei region and its surrounding areas: The role of primary pollutants and secondary aerosol formation, Sci. Total Environ., 849, 157621, https://doi.org/10.1016/j.scitotenv.2022.157621, 2022.

7. Lines 318 – 319: More discussion needs to be provided on how higher temperatures and more rainfall drive the distinct pH trends in Shanghai and Hong Kong.

**Response:** Thanks for your suggestion. We have added a discussion:

"Moreover, these coastal cities' warm climates amplify pH declines. Elevated temperatures reduce ALWC through moisture evaporation, concentrating $H^+$ and directly lowering pH. Concurrently, heat-enhanced $NH_3$ volatilization from particulate $NH_4^+$ weakened acid neutralization (Zhou et al., 2022; Nah et al., 2023)."

Nah, T., Lam, Y. H., Yang, J., and Yang, L.: Long-term trends and sensitivities of $PM_{2.5}$ pH and aerosol liquid water to chemical composition changes and meteorological parameters in Hong Kong, South China: Insights from 10-year records from three urban sites, Atmos. Environ., 302, https://doi.org/10.1016/j.atmosenv.2023.119725, 2023.

Zhou, M., Zheng, G., Wang, H., Qiao, L., Zhu, S., Huang, D., An, J., Lou, S., Tao, S., Wang, Q., Yan, R., Ma, Y., Chen, C., Cheng, Y., Su, H., and Huang, C.: Long-term trends and drivers of aerosol pH in eastern China, Atmos. Chem. Phys., 22, 13833–13844, https://doi.org/10.5194/acp-22-13833-2022, 2022.

Rainfall data has been removed from this study; therefore, the influence of rainfall on pH values will no longer be discussed.